# Explanation-Assisted Data Augmentation for Graph Learning

## Abstract

This work introduces a novel class of Data Augmentation (DA) techniques in the context of graph learning. In general, DA refers to techniques that enlarge the training set using label-preserving transformations. Such techniques enable increased robustness and generalization, especially when the size of the original training set is limited. A fundamental idea in DA is that labels are invariant to domain-specific transformations of the input samples. However, it is challenging to identify such transformations in learning over graphical input domains due to the complex nature of graphs and the need to preserve their structural and semantic properties. In this work, we propose explanation-assisted DA (EA-DA) for Graph Neural Networks (GNNs). A graph explanation is a subgraph which is an 'almost sufficient' statistic of the input graph with respect to its classification label. Consequently, the classification label is invariant, with high probability, to perturbations of graph edges not belonging to its explanation subgraph. We develop EA-DA techniques leveraging such perturbation invariances. First, we show analytically that the sample complexity of explanation-assisted learning can be arbitrarily smaller than explanation-agnostic learning. On the other hand, we show that if the training set is enlarged using EA-DA techniques and the learning rule does not distinguish between the augmented data and the original data, then the sample complexity can be worse than that of explanation-agnostic learning. We identify the main reason for the potential increase in sample complexity as the out-of-distribution nature of graph perturbations. We conclude that theoretically EA-DA may improve sample complexity, and that the learning rule must distinguish between the augmented data and the original data. Subsequently, we build upon these theoretical insights, introduce practically implementable EA-DA techniques and associated learning mechanisms, and perform extensive empirical evaluations.

## 1 Introduction

Graphs are used to represent relationships between entities in a wide range of applications including social networks, biology, and finance (Koller & Friedman, 2009; Barabási & Albert, 1999; de Dios Or-túzar & Willumsen, 2011; Barabási et al., 2011; Newman, 2018). In order to effectively leverage the rich relational information encoded in graphs, and inspired by conventional deep learning methods, various graph neural network (GNN) architectures have been developed, such as methods based on convolutional neural networks (Defferrard et al., 2016; Kipf & Welling, 2017), recurrent neural networks (Li et al., 2016; Ruiz et al., 2020), and transformers (Yun et al., 2019; Rong et al., 2020). Given the vast potential applications and use cases of GNNs, there is significant interest in developing data augmentation (DA) techniques to enhance their generalization capabilities and avoid overfitting during training (Kong et al., 2020; Han et al., 2022; Ling et al., 2023; Zhao et al., 2021; Rong et al., 2019).

In general, DA refers to techniques that enlarge the training set through label-preserving transformations. These techniques enhance generalization, especially when the size of the original training set is limited (Ding et al., 2022). A fundamental idea in DA is that labels are invariant to domain-specific transformations. For instance, in many image classification tasks, it is expected that the output label remains invariant to specific affine transformations of the original image, such as rotation and scaling. Thus, the training set can be enlarged using artificially generated samples created through these transformations. Building on the DA techniques used in non-graphical domains, techniques such

as Mixup (Han et al., 2022) and DropEdge (Rong et al., 2019) have been proposed for learning over graphs. However, in contrast to DA in non-graphical domains, in graphs even slight edge perturbations often lead to out-of-distribution samples. For instance, in molecular structures which are modeled as graphs, any edge perturbation that connects a carbon atom to more than four other atoms yields an out-of-distribution sample. Furthermore, classification labels are highly sensitive to edge modifications, and a single edge removal or addition may significantly change the properties of the molecular structure. As a result, it is challenging to identify label-preserving transformations in learning over graphs due to the complex nature of graphs and the need to preserve their structural and semantic properties and to ensure in-distribution augmentations. Moreover, it has been shown in learning over non-graphical domains that out-of-distribution augmentations can even lead to increased sample complexity (Shao et al., 2022).

In this work, we propose explanation-assisted data augmentation (EA-DA) for learning over graph-structured inputs. We introduce DA techniques that leverage the notion of subgraph explainability to enlarge the training set via label-preserving graph perturbations. This is based on the intuitive assumption that the presence of certain structural patterns or motifs within the input graph plays a critical role in the model's decision-making process (Ying et al., 2019; Luo et al., 2020; Yuan et al., 2021; Shan et al., 2021). Consequently, slight perturbations of the edges in the 'non-explanation' subgraph must be label-preserving. This assumption has been widely adopted in the literature of explainable GNNs (Ying et al., 2019; Luo et al., 2020; Yuan et al., 2022; Zheng et al., 2023). The label invariance to perturbations of non-explanation edges resembles the transformation invariances observed in various learning tasks on non-graphical data, such as invariance to scaling and rotation in image classification tasks (Cohen & Welling, 2016; Bloem-Reddy et al., 2020; Chen et al., 2020; Shao et al., 2022). To leverage this, we consider learning scenarios where each training sample, in addition to its associated label, is accompanied by its ground-truth explanation subgraph. Such ground-truth explanations may be produced at the time the training data is compiled. For example, in a dataset of labeled radiology scans, the most informative sections of each scan could be identified by the contributing physicians during the compilation phase of the training dataset. Alternatively, an estimate of the explanation can be produced by joint training of the classifier and its explainer on the original (unexplained) training data, as shown in the sequel. Consequently, we introduce explanation-assisted learning rules and data augmentation methods.

Our main contributions are summarized as follows:

- To provide a rigorous theoretical formulation of EA-DA mechanisms, the explanation-assisted graph learning problem, and the associated sample complexity.
- To introduce the explanation-assisted empirical risk minimization (EA-ERM) learning rule and to derive an upper-bound to its sample complexity. (Theorem 5.4)
- To show that the EA-ERM sample complexity can be arbitrarily smaller than the (explanation-agnostic) ERM sample complexity. (Example 5.3)
- To provide a theoretical justification, along with an example, showing that if EA-DA is used without distinguishing between original and augmented samples, then the sample complexity may be worse compared to that of the explanation-agnostic learners. (Example 6.2)
- To provide an implementable class of EA-DA mechanisms by building on the insights gained from our theoretical analysis. (Section 7)
- To provide empirical simulations verifying the improved performance of the GNNs trained using the EA-DA mechanisms when the necessary conditions in our theoretical derivations are satisfied, and to provide empirical simulations illustrating potentially worse performance in scenarios not satisfying the necessary conditions. (Section 8)

## 2 RELATED WORK

**Explainable Graph Neural Networks.** Prior works have introduced various methods for extracting subgraph explanations using GNNs (Ying et al., 2019; Luo et al., 2020; Yuan et al., 2020; 2022; 2021; Lin et al., 2021; Wang & Shen, 2023; Miao et al., 2023; Fang et al., 2023a; Xie et al., 2022; Ma et al., 2022). Traditional methods, such as SA (Baldassarre & Azizpour, 2019) and Grad-CAM (Pope et al., 2019), use gradients to extract explanations. Model-agnostic methods include perturbation-based methods, surrogate methods, and generation-based methods. Perturbation-based methods, including GNNExplainer (Ying et al., 2019), PGExplainer (Luo et al., 2020), and ReFine (Wang et al., 2021a),

generate perturbations to determine which features and subgraph structures are important. Surrogate methods (Vu & Thai, 2020; Duval & Malliaros, 2021) use a surrogate model to approximate the local prediction and use this surrogate model to generate explanations. Generation-based methods (Yuan et al., 2020; Shan et al., 2021; Wang & Shen, 2023) adopt generative models to derive instance-level and model-level explanations.

**Data Augmentation.** Data augmentation is widely used in self-supervised learning (You et al., 2020; Zhu et al., 2020). A large class of graph augmentation methods can be categorized as rule-based (Wang et al., 2021b; Rong et al., 2019; Gasteiger et al., 2019; Zhao et al., 2022a), learning-based methods (Zhao et al., 2021; Wu et al., 2022; Zhao et al., 2022b), and explanation-assisted data augmentation methods (Gu et al., 2023; Kwon & Lee, 2023; Wickramanayake et al., 2021; Tětková & Hansen, 2023; Shi et al., 2023).

Rule-based methods include NodeDrop (Rong et al., 2019), EdgeDrop (Feng et al., 2020), and MessageDrop (Fang et al., 2023b), which randomly drop a subset of features in the original graph. GraphCrop (Wang et al., 2020) and MoCL (Sun et al., 2021) randomly crop and substitute the graphs. Learning-based methods use GNNs to learn edge importance. For instance, ProGNN (Jin et al., 2020) learns a structural graph from a poisoned graph. GraphAug (Luo et al., 2022) introduces a reinforcement learning method to produce the label-invariant augmentations. Half-Hop (Azabou et al., 2023) proposes a novel graph augmentations by inserting a slow node. In (Liu et al., 2022), a local augmentation is proposed by learning the conditional distribution of the node under its neighbors.

EA-DA methods construct label-preserving transformations based on explanations. For instance, in (Gu et al., 2023), given the ground-truth explanation, a generative adversarial network (GAN) is used to generate image augmentations conditioned on the explanation sub-image. Other EA-DA methods (also called explanation-guided DA) have been studied recently (Gao et al., 2024), including in contrastive learning for sequential recommendation (Wang et al., 2022), image classification (Wickramanayake et al., 2021), and security analysis and risk detection (He et al., 2023). Mixup (Zhang et al., 2017) is a common strategy to generate explanation-assisted augmentations. In (Kwon & Lee, 2023), it is claimed that Mixup doesn't reflect the importance of each token in natural language processing, and a soft label assignment method is proposed. In the graph learning domain, (Shi et al., 2023) proposed a framework, ENGAGE, to use explanations to enhance contrastive learning representations.

## 3 PRELIMINARIES

### 3.1 THE GRAPH CLASSIFICATION PROBLEM

A graph $G$ is parametrized by i) a vertex[1] set $\mathcal{V} = \{v_1, v_2, \cdots, v_n\}$, where $n \in \mathbb{N}$, ii) an edge set $\mathcal{E} \subseteq \mathcal{V} \times \mathcal{V}$, iii) a feature matrix $\boldsymbol{X} \in \mathbb{R}^{n \times d}$, where the $i$th row $\boldsymbol{X}_i$ is associated with $v_i$ and $d$ is the feature dimension, and iv) an adjacency matrix $\boldsymbol{A} \in \{0, 1\}^{n \times n}$, where $A_{i,j} = \mathbb{1}((v_i, v_j) \in \mathcal{E})$. The graph is associated with a label $Y \in \mathcal{Y}$, where $\mathcal{Y}$ is a finite set. The graph parameters $(\boldsymbol{A}, \boldsymbol{X})$ and label $Y$ are generated based on the joint distribution $P_{Y, \boldsymbol{A}, \boldsymbol{X}}$. The notation $P_{Y,G}$ and $P_{Y, \boldsymbol{A}, \boldsymbol{X}}$ are used interchangeably. A classification scenario is completely characterized by $P_{Y,G}$; consequently, we refer to $P_{Y,G}$ as *the classification problem*. A graph classifier is a function $f : \mathcal{G} \to \mathcal{Y}$, where $\mathcal{G}$ is the support of $P_G$. Given $\epsilon \in [0, 1]$, the classifier is called $\epsilon$-accurate if $P(f(G) \neq Y) \leq \epsilon$.

A training set $\mathcal{T}$ is a collection of labeled graphs. The elements of the training set are generated independently and according to $P_{Y,G}$. A learning rule is a procedure that takes the training set $\mathcal{T}$ as input, and outputs a graph classifier $f(\cdot)$ belonging to an underlying hypothesis class $\mathcal{H}$.

### 3.2 SUBGRAPH EXPLANATIONS

At a high level, for a given task, an explanation function (explainer) $\Psi(\cdot)$ map the input graph $G$ to an explanation subgraph $G_{exp}$. The subgraph is a *good* explanation if it is *minimal* and *sufficient* with respect to $G$. The notions of minimality and sufficiency are rigorously quantified in the following.

The minimality of the subgraph is measured in terms of its number of edges (size). That is, $\Psi(G)$ is minimal if $\mathbb{E}(|\Psi(G)|)$ is as small as possible. Sufficiency means that the posterior distribution of the

---

[1]We use node and vertex interchangeably.

label $Y$ does not change significantly if we are given that $\Psi(G)$ is a subgraph of $G$ instead of the complete realization of $G$. That is, the explanation subgraph is sufficient if $d_{TV}(P_{Y|G=g}, P_{Y|\Psi(g)\subseteq G})$ is small for all $g \in \mathcal{G}$, where $d_{TV}$ denotes the total variation distance. Consequently, for given parameters $s \in \mathbb{N}$ and $\kappa > 0$, we say that the mapping $\Psi(\cdot)$ is an $(s, \kappa)$-explainer for the task $P_{Y,G}$ if:

$$d_{TV}(P_{Y|G=g}, P_{Y|\Psi(g)\subseteq G}) \leq \kappa, \forall g \in \mathcal{G} \quad \text{and} \quad \mathbb{E}(|\Psi(G)|) \leq s. \tag{1}$$

If an $(s, \kappa)$-explainer exists, we say that the task is $(s, \kappa)$-explainable.

Note that for any $\kappa \geq 0$ and any given classification task $P_{Y,G}$, the task is trivially $(\mathbb{E}(|G|), \kappa)$-explainable since the graph itself can be taken as its explanation, i.e., $\Psi(G) = G$. Furthermore, in most practical scenarios, input graphs contain redundant edges, and consequently, the tasks are $(s, \kappa)$-explainable for an $s$ which is strictly smaller than $\mathbb{E}(|G|)$. In the subsequent sections, we leverage such redundancies to design EA-DA methods for graph learning.

### 3.3 EXPLANATION-ASSISTED LEARNING RULES

As described in the introduction, in our theoretical analysis, we consider learning rules that jointly operate on labeled training samples and their associated subgraph explanations. Formally, given a hypothesis class $\mathcal{H}$, an explanation-assisted learning rule is a mapping $L_{EA} : (\mathcal{T}, \Psi_{|\mathcal{T}}(\cdot)) \mapsto f(\cdot)$, where $\mathcal{T}$ is the training set, $\Psi(\cdot)$ is an explainer, and $\Psi_{|\mathcal{T}}(\cdot)$ is its restriction to the training set[2]. The sample complexity of explanation-assisted learning rules is defined as follows.

**Definition 3.1 (Explanation-Assisted Sample Complexity).** *Let $\epsilon, \delta, \kappa, \gamma \in (0, 1)$. The sample complexity of $(\epsilon, \delta, \kappa, \gamma)$-PAC learning of $\mathcal{H}$ with respect to the explanation function $\Psi(\cdot)$, denoted by $m_{EA}(\epsilon, \delta, \kappa, \gamma; \mathcal{H}, \Psi)$, is defined as the smallest number of training samples $m \in \mathbb{N}$ for which there exists an explanation-assisted learning rule $L$ such that, for every $s \in \mathbb{N}$ and $(s, \kappa)$-explainable task $P_{Y,G}$ with Bayes error rate less than or equal to $\gamma$, we have:*

$$P\left(err_{P_{Y,G}}(L(\mathcal{T})) \leq \inf_{f \in \mathcal{H}} err_{P_{Y,G}}(f) + \epsilon\right) \geq 1 - \delta,$$

*where we have defined $err_{P_{Y,G}}(f)$ as the statistical error of $f(\cdot)$ for the task $P_{Y,G}$, and $|\mathcal{T}| = m$. If no such $m$ exists, then we say the sample complexity is infinite.*

Note that in addition to the parameters $(\epsilon, \delta)$ used in the standard PAC formulation, and the explainability parameter $\kappa$, the sample complexity is parameterized by an upper-bound on the Bayes error rate $\gamma$. If $\gamma = 1$, we recover the agnostic PAC settings; if $\gamma = 0$ the task is deterministic, and if the optimal (zero-error) classifier is in the hypothesis class, we recover the realizable PAC settings. The explicit dependence on $\gamma$ is needed to derive the bounds on EA-DA sample complexity in the sequel.

## 4 EXPLAINABLE TASKS AND PERTURBATION-INVARIANCE

The fundamental idea in DA techniques is that in many application domains, there are label-preserving transformations that can be applied to enlarge the training set and facilitate generalization. We argue that for certain classes of graph learning tasks, graph transformations that only alter the non-explanation subgraphs are label-preserving with high probability. To elaborate, let us consider a task with small Bayes error rate, so that the input graph $G$ accurately captures the label $Y$. Then, if the task is explainable, from equation 1 it follows that for two input graphs $G$ and $G'$, if the explanation $\Psi(G)$ is a subgraph of $G'$, then $G$ and $G'$ have the same label, with high probability. Thus, such (almost) label-preserving transformations can be used for EA-DA. It should be noted that the label-preserving property depends on the Bayes error rate, and if the error rate is high, then such transformations may not be label-preserving. The relationship between the Bayes error rate, explainability, and perturbation invariance is formally quantified in the following proposition.

**Proposition 4.1 (Perturbation Invariance and Explainability).** *Let $\kappa, \gamma \geq 0$ such that $\gamma + 2\kappa \leq 1$ and let $s \in \mathbb{N}$. Then, for any $(\kappa, s)$-explainable task $P_{Y,G}$ with Bayes error $\gamma$, the following holds:*

$$\sum_{g_{exp}} P(\Psi(G) = g_{exp})P(Y \neq Y'|\Psi(G) = g_{exp}, \quad g_{exp} \subseteq G') \leq -\gamma^2 - 2\kappa^2 + 2\gamma + 3\kappa - 3\gamma\kappa.$$

---

[2]There is a slight abuse of notation as the domain of $\Psi(\cdot)$ is restricted to the graphs samples in the training set, however, we denote the restriction by $\Psi_{|\mathcal{T}}(\cdot)$ to avoid unnecessary introduction of new notation.

where $\Psi(\cdot)$ is a $(\kappa, s)$-explanation function for $P_{Y,G}$, $Y$ and $Y'$ are the labels associated with $G$ and $G'$, respectively, and $(G, Y)$ and $(G', Y')$ are generated independently and according to $P_{Y,G}$.

The proof follows directly from the definition of explainability in equation 1 (Appendix A.1).

## 5 PAC Learnability of Explanation-Assisted Learners

The notion of perturbation invariance is analogous to transformation invariances, such as rotation and scaling invariances, observed in image classification. Prior works on sequential data have shown that invariance-aware learning rules can achieve improved sample complexity, e.g., (Shao et al., 2022). Building on this, we introduce the explanation-assisted ERM (EA-ERM) and derive an upper-bound on its sample complexity. We provide an example where this sample complexity can be arbitrarily smaller than that of (explanation-agnostic) ERM. We conclude that for explainable classification tasks, there may be significant benefits in using explanation-assisted learning rules, in terms of sample complexity. This is further verified via empirical analysis in the subsequent sections. It should be noted that while our observations in this section regarding the improved sample complexity of EA-DA methods align with those of prior works including (Shao et al., 2022), and the proof techniques build upon prior known methods, there are several crucial differences which merit a separate treatment. First, the label-preserving transformation considered in prior works, such as rotations and color translations of images, form closed groups, which facilitate analysis. In contrast, the transformations considered in this work, which include graph perturbations by addition and omission of edges in the non-explanation subgraph, do not form closed groups. Second, the transformations considered in prior works are assumed to be completely label-preserving, whereas the graph perturbation operations considered in this work are almost label-preserving and probabilistic. This introduces new challenges in evaluating the resulting sample complexity, which are addressed in the following sections.

**Definition 5.1** (**Explanation-Assisted ERM (EA-ERM)**). *Given a hypothesis class $\mathcal{H}$, training set $\mathcal{T}$, and explanation function $\Psi(\cdot)$, the EA-ERM learning rule produces $L_{\text{EA-ERM}}(\mathcal{T}) \triangleq \widetilde{f}(\cdot)$, where:*

$$\widetilde{f}(G) \triangleq \begin{cases} Y_{exp} & \exists i \in [t] : \Psi(G_i) \subseteq G, \\ f(G) & Otherwise \end{cases}, \qquad f(\cdot) \triangleq L_{ERM}(\mathcal{T}), \qquad (2)$$

*where $Y_{exp}$ is chosen randomly and uniformly from the set $\{Y_i | \Psi(G_i) \subseteq G, i \in |\mathcal{T}|\}$, and $L_{ERM}$ denotes the (explanation-agnostic) ERM learning rule.*

Note that Definition 5.1 implies a two-step learning procedure. First, given a training set $\mathcal{T}$, a classifier $f(\cdot)$ is trained by applying the ERM learning rule $L_{ERM}$. Then, $\widetilde{f}(\cdot)$ is constructed from $f(\cdot)$ using equation 2. We will show that the sample complexity of EA-ERM can be expressed in terms of the explanation-assisted VC dimension defined in the following.

**Definition 5.2** (**Explanation-Assisted VC Dimension**). *Given an explanation function $\Psi(\cdot)$ and hypothesis class $\mathcal{H}$, the explanation-assisted VC dimension $VC_{EA}(\mathcal{H}, \Psi)$ is defined as the largest integer $k$ for which there exists a collection of graphs $\mathcal{G} = \{g_1, g_2, \cdots, g_k\}$ such that $\Psi(g_i) \neq \Psi(g_j)$ for all $i \neq j$, and every labeling of $\mathcal{G}$ is realized by the hypothesis class $\mathcal{H}$.*

Let us define $\mathbf{I}(G) = G$ as the identity function. We call $VC(\mathcal{H}) \triangleq VC_{EA}(\mathcal{H}, \mathbf{I})$ the *standard* VC dimension, as it aligns with the notion of VC dimension considered in traditional PAC learnability analysis. The following provides a simple example in which the standard VC dimension, $VC(\mathcal{H})$, can be arbitrarily larger than the explanation-assisted VC dimension $VC_{EA}(\mathcal{H}, \Psi(\cdot))$.

**Example 5.3.** *Let $C_i, i \in \mathbb{N}$ denote the single-cycle graph with $i$ vertices, where the vertex set is[3] $\mathcal{V} = [i]$ and the edge set is $\mathcal{E} = \{(j, j + 1), j \in [i - 1]\} \cup \{(i, 1)\}$. We construct a binary classification problem as follows. Let the graphs associated with label zero belong to the collection $\mathcal{B}_0 = \{C_i \cup C_3, i > 5\}$ and those associated with label one belong to $\mathcal{B}_1 = \{C_i \cup C_4, i > 5\}$. Let*

$$\Psi(G) = \begin{cases} C_3 & if\ G \in \mathcal{B}_0 \\ C_4 & otherwise \end{cases}.$$

*Clearly, $\kappa = 0$. Let $P_Y(0) = P_Y(1) = \frac{1}{2}$, and assume that for a given label $Y = y$, the graphs belonging to $\mathcal{B}_y$ are equally likely, i.e., $P_{G|Y}(\cdot|y)$ is uniform. Let $\mathcal{H}$ consist of all possible classifiers*

---
[3]For conciseness, we denote the set $\{1, 2, \cdots, i\}$ by $[i]$.

on the set $\mathcal{B}_0 \cup \mathcal{B}_1$. So that $VC(\mathcal{H}) = \infty$. It is straightforward to see that $VC_{EA}(\mathcal{H}, \Psi(\cdot)) = 2$ since there are only two explanation graphs, namely $C_3$ and $C_4$.

Next, we show that the sample complexity of explanation-assisted learning rules is expressed in terms of $VC_{EA}(\mathcal{H}, \Psi(\cdot))$ as opposed to $VC(\mathcal{H})$ achieved by generic learning rules.

**Theorem 5.4 (Sample Complexity of Explainable Tasks).** *Let $\epsilon, \delta, \kappa, \gamma \in (0, 1)$ such that*

$$-\gamma^2 - 2\kappa^2 + 2\gamma + 3\kappa - 3\gamma\kappa \leq \frac{\epsilon}{32},$$

*then, for any hypothesis class $\mathcal{H}$ and explanation function $\Psi(\cdot)$, the following holds:*

$$m_{EA}(\epsilon, \delta, \kappa, \gamma; \mathcal{H}, \Psi) = O\left(\frac{d}{\epsilon^2} \log^2 d + \frac{1}{\epsilon^2} ln(\frac{1}{\delta})\right),$$

*where we have defined $d \triangleq VC_{EA}(\mathcal{H}, \Psi(\cdot))$.*

The proof is provided in Appendix A.2. An important implication of the proof steps is that EA learning rules may significantly improve sample complexity if the Bayes error rate of the task is small enough. In the subsequent sections, we show that the learning rule should distinguish between the original training data and its EA perturbations to achieve the potential improvements.

## 6 PAC LEARNABILITY OF EXPLANATION-ASSISTED DATA AUGMENTATION

In the previous section, we showed that explanations can potentially be leveraged to improve sample complexity. One method for utilizing the explanation subgraphs is to perform EA-DA, by artificially producing training inputs via edge additions and omissions in the non-explanation subgraph. In this section, we show through a simple example that this approach may lead to worse sample complexity compared to generic explanation-agnostic learning rules if the learning rule does not distinguish between the original data and the augmented data. This observation aligns with recent observations in (Shao et al., 2022) in the context of other transformation invariances such as rotations and scalings. The phenomenon is also observed in our empirical observations in the subsequent sections.

**Definition 6.1 (Explanation-Preserving Perturbation).** *Consider a task $P_{Y,G}$, an explainer $\Psi(\cdot)$, and a parameter $\alpha > 0$. An explanation-preserving perturbation $S^\alpha(G)$ is a mapping* [4]

$$S^\alpha(G) \triangleq \{G' \big| \Psi(G) \subseteq G', |\mathcal{E}\Delta\mathcal{E}'| \leq \alpha|\mathcal{E}|\},$$

*where $\mathcal{E}$ and $\mathcal{E}'$ are the edge sets of $G$ and $G'$, respectively, and $\Delta$ denotes the symmetric difference.*

Given training set $\mathcal{T}$, explainer $\Psi(\cdot)$, and $\alpha > 0$, we define the EA augmented training set as:

$$\mathcal{T}_{aug} \triangleq \mathcal{T} \cup \Big( \bigcup_{(G,Y) \in \mathcal{T}} \{(G', Y) | G' \in S^\alpha(G)\} \Big).$$

We define the DA-ERM learning rule as an ERM learning rule that is applied to the augmented training set without distinguishing between the original and augmented data. For $\alpha = 0$, DA-ERM is the same as ERM and there is no data augmentation. The following example shows that in general, for $\alpha > 0$, DA-ERM may have worse sample complexity than the explanation-agnostic ERM.

**Example 6.2.** *Consider the hypothesis class $\mathcal{H}$ which consists of all classifiers that classify their input only based on the number of edges in the graph. That is,*

$$\mathcal{H} = \{f(\cdot) | \forall G, G' : |G| = |G'| \rightarrow f(G) = f(G')\}.$$

*Furthermore, let us consider the following binary classification problem. Let $P_Y(0) = P_Y(1) = \frac{1}{2}$, and let the graphs associated with label zero consist of the collection*

$$\mathcal{B}_0 = \{G \big| |G| = n, \exists i \in [n] : C_i \subseteq G \text{ and } \nexists j \subseteq [n] : D_j \subseteq G\},$$

*where $n > 10$ is fixed, $C_i, i \in [n]$ denotes a cycle of size $i$, and $D_i$ denotes a star of size $i$, where a star is a subgraph where all vertices are connected to a specific vertex called the center, and there*

---

[4] $S^\alpha(G)$ is defined with respect to $\Psi(\cdot)$. This dependence is not made explicit in our notation to avoid clutter.

*are no edges between the rest of the vertices. Thus, $\mathcal{B}_0$ consists of all graphs with exactly $n$ edges and at least one cycle but no stars. Similarly, let the graphs associated with label one be given by*

$$\mathcal{B}_0 = \{G \big| |G| = n + 1, \not\exists i \in [n+1] : C_i \subseteq G \text{ and } \exists j \in [n+1] : D_j \subseteq G\}.$$

*That is, $\mathcal{B}_1$ consists of all graphs with exactly $n + 1$ edges that do not contain a cycle but contain a star. Let $\alpha = \frac{1}{n}$, and define*

$$\Psi(G) \triangleq \begin{cases} C_i & \text{if } \exists i : C_i \subseteq G, \\ D_i & \text{if } \exists i : D_i \subseteq G, \end{cases}.$$

*Clearly $\kappa = \gamma = 0$. It is straightforward to see that ERM and EA-ERM both achieve zero error after observing at least one sample per label since all graphs of size $n$ have label 0 and all graphs of size $n + 1$ have label 1, and the hypothesis class decides based only on the number of edges. On the other hand, for DA-ERM to achieve zero error it needs to observes all possible explanation outputs, as it cannot distinguish between the augmented elements of $\mathcal{B}_0$ and the original elements of $\mathcal{B}_1$ and vice versa since they may have the same number of edges. Thus, data augmentation has sample complexity that can grow arbitrarily large, whereas ERM and EA-ERM have sample complexity equal to two.*

The issue illustrated in the previous example appears to be a fundamental issue. To explain further, note that DA-ERM empirically minimizes the risk over the augmented dataset. If the elements of the augmented dataset are in-distribution with respect to $P_{Y,G}$, this also guarantees that the risk is minimized with respect to the original dataset, hence achieving similar performance as that of EA-ERM. However, if the elements of the augmented dataset are out-of-distribution with respect to $P_G$, then it may be the case that the output of DA-ERM performs well on the out-of-distribution elements but has high error on the in-distribution elements (which are dominated by the out-of-distribution elements). Hence, DA-ERM may achieve high error probability on the original data distribution. This is exactly the phenomenon observed in the previous example. We show this phenomenon empirically and further explain it in our empirical evaluations in the subsequent sections.

# 7 EXPLANATION-ASSISTED GNN ARCHITECTURES

In this section, we introduce a practically implementable EA-DA method and GNN training procedure. Given a labeled training sample $(G, Y)$ and explanation function $\Psi(\cdot)$, we first compute an explanation subgraph $G_{exp} = \Psi(G)$. Then, we use an explanation-preserving, non-parametric perturbation operator $\Pi(\cdot)$ to produce perturbations $G_i$ of the original input graph $G$, such that $G_{exp} \subseteq G_i$. Then, $G$ and $G_i$ are passed through the GNN $f(\cdot)$ to produce the output labels $\widehat{Y}$ and $\widehat{Y}_i$, respectively.

For a fixed parameter $\lambda > 0$, the loss is defined as:

$$Loss = CE(Y, \widehat{Y}) + \lambda \sum_i CE(Y, \widehat{Y}_i). \tag{3}$$

As shown in Section 6, if the perturbed graphs are out-of-distribution, then the performance may be worse than explanation-agnostic methods. To address this, first, we follow an existing work to implement the perturbation function $\Pi(\cdot)$ which randomly removes a small number of non-explanation edges (Zheng et al., 2023) (Algorithm 2). As shown in previous studies, this method is effective in generating in-distribution graphs. Second, to further alleviate the negative effects of out-of-distributed augmentations, we choose the hyperparameter $\lambda$ (in Eq. 3) small enough, so that the loss on the (potentially out-of-distribution) augmented data does not dominate the loss on the original data.

**Algorithm 1** Explanation-Assisted Training Algorithm

1: **Input:** Training set $\mathcal{T}$, balancing coefficient $\lambda$, GNN pre-train epoch $e_w$, train epoch $e_s$, sampling number $M$
2: **Output:** Trained model $f$
3: Initiate $f$, $\Psi$, $j = 0$
4: **for** $j \leq e_w$ **do**
5:     Update $f$ via $\mathbb{E}_{\mathcal{T}}(CE(Y, f(G)))$
6:     $j = j + 1$
7: **end for**
8: Train the explainer $\Psi(\cdot)$ on $\mathcal{T}$
9: Initiate empty set $\mathcal{T}'$
10: **for** each $(G, Y) \in \mathcal{T}$ **do**
11:     $G_{exp} = \Psi(G)$
12:     **for** $m$ in $[1, 2, ...M]$ **do**
13:         $\mathcal{T}' = \mathcal{T}' + \{(\Pi(G_{exp}), Y)\}$.
14:     **end for**
15: **end for**
16: Initialize $f$, $j = 0$
17: **for** $j \leq e_s$ **do**
18:     Train $f$ with $\mathbb{E}_{\mathcal{T}}(CE(Y, f(G)) + \lambda\mathbb{E}_{\mathcal{T}'}CE(Y, f(G)))$
19:     $j = j + 1$
20: **end for**

Table 1: Performance comparisons with 3-layer GNNs trained on 50 samples. The metric is classification accuracy. The best results are shown in bold font and the second best ones are underlined.

| Method | MUTAG | | Benzene | | Fluoride | | Alkane | | D&D | | PROTEINS | |
|---|---|---|---|---|---|---|---|---|---|---|---|---|
| | GCN | GIN | GCN | GIN | GCN | GIN | GCN | GIN | GCN | GIN | GCN | GIN |
| Vanilla | 84.3±3.2 | 82.5±3.7 | 73.9±5.2 | 67.5±5.9 | 62.1±4.3 | 68.6±5.2 | 93.7±3.2 | 85.1±10.3 | 63.2±6.6 | 65.1±4.3 | 68.1±6.1 | 66.5±4.0 |
| EI | 85.6±2.0 | 82.8±3.2 | 75.3±5.1 | 71.6±2.8 | 59.3±3.2 | 66.8±4.0 | 92.2±4.9 | 87.5±10.3 | 63.6±6.2 | 64.7±5.4 | 69.6±4.0 | 65.5±5.8 |
| ED | 84.7±3.4 | 81.6±3.7 | 73.2±4.1 | 70.5±4.3 | 58.4±3.7 | 62.9±5.1 | 94.4±1.8 | 90.3±6.4 | 64.0±6.0 | 66.7±3.8 | 70.0±4.0 | 62.7±5.3 |
| ND | 83.6±3.5 | 82.2±4.0 | 74.0±3.8 | 71.3±2.7 | 58.7±3.0 | 64.9±4.6 | 92.7±3.4 | 88.9±7.0 | 65.2±4.2 | 66.7±2.9 | 68.8±3.3 | 65.6±5.4 |
| FD | 84.7±3.4 | 82.7±2.9 | 75.2±4.8 | 70.7±2.8 | 57.6±3.6 | 67.6±5.1 | 93.8±3.1 | 83.1±11.7 | 62.5±3.3 | 68.2±4.3 | 68.6±3.8 | 65.6±5.0 |
| Mixup | 67.4±3.2 | 74.5±1.6 | 53.9±1.9 | 59.0±3.4 | 52.5±1.5 | 51.6±2.6 | 64.3±0.7 | 65.8±4.1 | 56.0±1.9 | 58.6±3.5 | 60.8±2.9 | 62.2±2.9 |
| Aug$_{GE}$ | 87.2±1.4 | 86.0±2.4 | 76.2±1.3 | **75.4±0.8** | **66.6±3.4** | 76.3±2.1 | 96.3±1.3 | **94.9±1.1** | 66.1±5.1 | **69.3±5.2** | 70.4±5.9 | **68.5±5.9** |
| Aug$_{PE}$ | **87.2±2.6** | **86.9±1.8** | **76.5±0.8** | 75.4±1.0 | 65.3±5.0 | **76.5±1.7** | **96.4±1.1** | 94.8±1.1 | **67.7±4.3** | 67.4±2.8 | **71.2±6.3** | 68.1±5.5 |

It should be noted that the ground-truth explanation $\Psi(G)$ may not be available beforehand in real-world applications. In such scenarios, we pre-train the graph classifier $f(\cdot)$ and $\Psi(\cdot)$. This two-step training procedure is described in Algorithm 1. The proposed method is a general framework that can be employed for training various GNN architectures and explainers, such as GIN (Xu et al., 2019), PNA (Corso et al., 2020), GNNExplaier (Ying et al., 2019) and PGExplainer (Luo et al., 2020).

# 8 EMPIRICAL VERIFICATION

We utilize a benchmark synthetic dataset, BA-2motifs (Luo et al., 2020), and five real-world datasets, MUTAG (Luo et al., 2020), Benzene, Fluoride, Alkane (Agarwal et al., 2023), D&D (Dobson & Doig, 2003) and PROTEINS (Dobson & Doig, 2003;

---

**Algorithm 2** Explanation-preserving perturbation $\Pi(\cdot)$

1: **Input:** a graph $G$, explainer $\Psi(\cdot)$, hyper-parameter $\alpha_1$.
2: $G^c = G - \Psi(G)$       # Compute the non-explanation subgraph
3: $E_{\alpha_1}(G^c) =$ sample $\alpha_1$ edges from $G^c$
4: **Return** $E_{\alpha_1}(G^c) + \Psi(G)$

---

Borgwardt et al., 2005). We consider four GNN models: Graph Convolutional Network (GCN), Graph Isomorphism Network (GIN), Principal Neighbourhood Aggregation (PNA) (Corso et al., 2020) and GraphSage(Hamilton et al., 2017). Full experimental setups are shown in the Appendix B.

## 8.1 COMPARISON TO BASELINE DATA AUGMENTATIONS

With this set of experiments, we aim to verify the effectiveness of our EA-DA methods.

**Experiment Design.** We consider 3 GNN layers. For each dataset, 50 labeled graphs are randomly sampled for training and 10% of the graphs for testing. Experiments with smaller training sizes and lightweight GNN models can be found in Appendix C.5 and C.6, respectively. We compare with representative structure-oriented augmentations, Edge Inserting (EI), Edge Dropping (ED), Node Dropping (ND), and Feature Dropping(FD) (Ding et al., 2022). Recently, mixup operations have been introduced in the graph domain for DA, such as $M$-mixup (Wang et al., 2021b) and $G$-mixup (Han et al., 2022). However, $M$-mixup operates on the embedding space and cannot be fairly compared, and $G$-mixup does not apply to graphs with node type/features. Instead, we use a normal Mixup as an additional baseline. To generate $\Psi(\cdot)$ in our method, we consider two representative explainers, GNNExplainer (Ying et al., 2019) and PGExplainer (Luo et al., 2020), whose corresponding augmentations are denoted by Aug$_{GE}$ and Aug$_{PE}$, respectively. More comprehensive results on different settings and full experimental results are shown in Appendix C.

**Experimental Results.** From Tables 1 and 3 (in the Appendix), we have the following observations. First, our explanation-assisted learning methods consistently outperform the vanilla GNN models as well as the ones trained with structure-oriented augmentations by large margins. Utilizing the GCN as the backbone, our methods, Aug$_{GE}$ and Aug$_{PE}$, exhibit significant enhancements in classification accuracy—2.40% and 2.75%, on average—when compared to the best-performing baselines across six datasets. With GIN, the improvements are 5.04% and 4.69%, respectively. Secondly, we observe that traditional structure-based augmentation methods yield comparatively less effectiveness. For example, in the Fluoride dataset, all baseline augmentation methods achieve negative effects, while our methods can still beat the backbone significantly.

## 8.2 EFFECTS OF AUGMENTATION DISTRIBUTION

In Section 6, we investigated EA-DA, and argued that performance improvements are contingent on in-distribution generation of augmented data. In this section, we analyze the effects of augmentation distributions on the model accuracy. With this set of experiments on a synthetic dataset and a real-world dataset, we aim to explore two questions: (RQ1) Can in-distribution augmentations lead to better data efficiency in graph learning? (RQ2) What are the effects of out-of-distribution (OOD) augmentations on graph learning?

To evaluate the data efficiency of graph learning methods, we vary the number of training samples in the range $[4, 8, 20, 40, 100, 300, 500, 700]$. We sufficiently train GNN models with three settings: 1) training with the vanilla training samples, 2) training with in-distribution EA-DA, and 3) training with OOD EA-DA. For setting 2, we use the proposed augmentation method on the ground truth explanations. For setting 3, to generate OOD

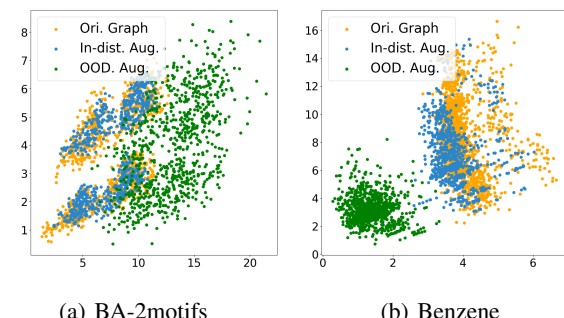

(a) BA-2motifs        (b) Benzene

Figure 1: Original input, in-distribution, and OOD augmentation embeddings generated by T-SNE.

augmentations, we randomly add 100% edges from the BA graph for each instance on the BA-2motifs dataset. On the Benzene dataset, we randomly remove 30% edges from the non-explanation subgraphs. Visualization results on three sets of graphs are shown in Figure 1, which shows that our methods are able to generate both in-distributed and OOD augmentations for further analysis.

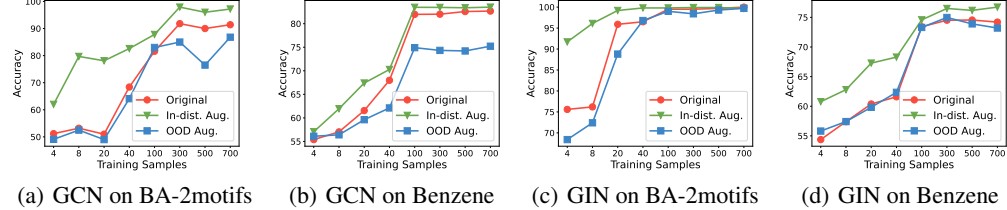

(a) GCN on BA-2motifs    (b) GCN on Benzene    (c) GIN on BA-2motifs    (d) GIN on Benzene

Figure 2: Effects of in-distributed and OOD augmentations on the accuracy of GCN and GIN on BA-2motifs and Benzene datasets.

We answer our research questions with accuracy performances in Figure 2. From these figures, we have the following observations. First, in-distribution augmentations significantly and consistently improve the data efficiency of both GCN and GIN in two datasets. For example, with explanation-preserving augmentations, GIN can achieve over 90% accuracy with only 4 samples in the synthetic dataset, while the performance of GIN trained with original datasets is around 75%. Second, OOD augmentations fail to improve data efficiency in most cases. Moreover, for GCN on Benzene, the OOD augmentation worsens the performance, which is aligned with our theoretical analysis.

## 9 CONCLUSION

Explanation-assisted learning rules were considered, where in additional to labeled training samples, the learning rule has access to explanation subgraphs. The sample complexity was characterized and was shown to be arbitrarily smaller than the explanation-agnostic sample complexity. Subsequently, explanation-assisted data augmentation methods were considered, followed by a generic learning rule applied to the augmented dataset. It was shown both theoretically and empirically that this may sometimes lead to better and sometimes to worse performance in terms of sample complexity, where gains are contingent on producing in-distribution augmented samples.

IMPACT STATEMENTS

This paper presents work whose goal is to advance the field of Machine Learning. There are many potential societal consequences of our work, none of which we feel must be specifically highlighted here.

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

# A PROOFS

## A.1 PROOF OF PROPOSITION 4.1

We have the following:

$$\sum_{g_{exp}} P(\Psi(G) = g_{exp}) P(Y \neq Y' | \Psi(G) = g_{exp}, \quad g_{exp} \subseteq G')$$

$$= \sum_{g_{exp}} P(\Psi(G) = g_{exp}) \sum_{\substack{y,y' \in \mathcal{Y} \\ y \neq y'}} P(Y = y, Y' = y' | \Psi(G) = g_{exp}, \quad g_{exp} \subseteq G')$$

$$\overset{(a)}{=} \sum_{g_{exp}} P(\Psi(G) = g_{exp}) \sum_{\substack{y,y' \in \mathcal{Y} \\ y \neq y'}} P(Y = y | \Psi(G) = g_{exp}) P(Y' = y' | g_{exp} \subseteq G')$$

$$\overset{(b)}{=} \sum_{g_{exp}} P(\Psi(G) = g_{exp}) \sum_{\substack{y,y' \in \mathcal{Y} \\ y \neq y'}} P(Y = y | \Psi(G) = g_{exp}) P(Y = y' | g_{exp} \subseteq G)$$

$$= \sum_{g_{exp}} P(\Psi(G) = g_{exp}) \sum_{y \in \mathcal{Y}} \left( P(Y = y | \Psi(G) = g_{exp}) \sum_{y' \neq y} P(Y = y' | g_{exp} \subseteq G) \right)$$

$$= \sum_{g_{exp}} P(\Psi(G) = g_{exp}) \sum_{y \in \mathcal{Y}} P(Y = y | \Psi(G) = g_{exp})(1 - P(Y = y | g_{exp} \subseteq G))$$

$$= 1 - \sum_{g_{exp}} P(\Psi(G) = g_{exp}) \sum_{y \in \mathcal{Y}} P(Y = y | \Psi(G) = g_{exp}) P(Y = y | g_{exp} \subseteq G)$$

where in (a) we have used the independence of $(G, Y)$ and $(G', Y')$, and in (b) we have used the fact that $(G, Y)$ and $(G', Y')$ are identically distributed. Furthermore:

$$\sum_{g_{exp}} P(\Psi(G) = g_{exp}) \sum_{y \in \mathcal{Y}} P(Y = y | \Psi(G) = g_{exp}) P(Y = y | g_{exp} \subseteq G)$$

$$= \sum_{g_{exp}} \left( \sum_{g} P(G = g, \Psi(G) = g_{exp}) \right) \sum_{y \in \mathcal{Y}} P(Y = y | \Psi(G) = g_{exp}) P(Y = y | g_{exp} \subseteq G)$$

$$= \sum_{g} \sum_{g_{exp}} P_G(g) \mathbb{1}(\Psi(g) = g_{exp}) \sum_{y \in \mathcal{Y}} P(Y = y | \Psi(G) = g_{exp}) P(Y = y | g_{exp} \subseteq G)$$

$$= \sum_{g} P_G(g) \sum_{y \in \mathcal{Y}} P(Y = y | \Psi(G) = \Psi(g)) P(Y = y | \Psi(g) \subseteq G)$$

$$\geq \sum_{g} P_G(g) P(Y = f^*(g) | \Psi(G) = \Psi(g)) P(Y = f^*(g) | \Psi(g) \subseteq G)$$

where $f^*(\cdot)$ denotes the Bayes decision rule and we have used the fact that probability is non-negative to remove the $y \neq f^*(g)$ terms in the summation over $y \in \mathcal{Y}$. Hence, we have[5]

$$\sum_{g_{exp}} P(\Psi(G) = g_{exp}) \sum_{y \in \mathcal{Y}} P(Y = y | \Psi(G) = g_{exp}) P(Y = y | g_{exp} \subseteq G)$$

$$\geq \sum_{g} P_G(g) P(Y = f^*(g) | \Psi(G) = \Psi(g)) P(Y = f^*(g) | \Psi(g) \subseteq G)$$

$$\overset{(a)}{\geq} \sum_{g} P_G(g) P(Y = f^*(g) | \Psi(G) = \Psi(g))(P(Y = f^*(g) | G = g) - \kappa) \tag{4}$$

---

[5]Note that we have defined total variation distance as $d_{TV}(P_G, Q_G) \triangleq \sum_{g} |P_G(g) - Q_G(g)|$. An alternative definition in some textbooks contains a factor of $\frac{1}{2}$.

where in (a) we have used equation 1 and the definition of total variation distance. On the other hand:

$$
\begin{aligned}
P(Y &= f^*(g)|\Psi(G) = \Psi(g)) \\
&= \sum_{g':\Psi(g')=\Psi(g)} P(Y = f^*(g), G = g'|\Psi(G) = \Psi(g)) \\
&= \sum_{g':\Psi(g')=\Psi(g)} P(G = g'|\Psi(G) = \Psi(g))P(Y = f^*(g)|G = g') \\
&\overset{(a)}{\geq} \sum_{g':\Psi(g')=\Psi(g)} P(G = g'|\Psi(G) = \Psi(g))(P(Y = f^*(g)|\Psi(g') \subseteq G) - \kappa) \\
&\overset{(b)}{\geq} \sum_{g':\Psi(g')=\Psi(g)} P(G = g'|\Psi(G) = \Psi(g))(P(Y = f^*(g)|G = g) - 2\kappa) \\
&\overset{(c)}{=} P(Y = f^*(g)|G = g) - 2\kappa
\end{aligned}
\tag{5}
$$

where in (a) we have used the fact that $G = g'$ implies that $\Psi(g') \subseteq G$ along with equation 1, in (b) we have used the fact that $\Psi(g') = \Psi(g)$ to conclude that $\Psi(g) \subseteq G$ and used equation 1, and in (c) we have used the law of total probability. Consequently, from equation 4 and equation 5, we have:

$$
\sum_{g_{exp}} P(\Psi(G) = g_{exp}) \sum_{y \in \mathcal{Y}} P(Y = y|\Psi(G) = g_{exp})P(Y = y|g_{exp} \subseteq G)
$$

$$
\begin{aligned}
&\geq \sum_g P_G(g)P(Y = f^*(g)|\Psi(G) = \Psi(g))(P(Y = f^*(g)|G = g) - \kappa) \\
&\geq \sum_g P_G(g)(P(Y = f^*(g)|G = g) - 2\kappa)(P(Y = f^*(g)|G = g) - \kappa) \\
&= \mathbb{E}_G((P(Y = f^*(G)) - 2\kappa)(P(Y = f^*(G)) - \kappa)) \\
&\geq (1 - \gamma - \kappa)(1 - \gamma - 2\kappa),
\end{aligned}
$$

where we have used the definition of the Bayes error rate and the assumption that $1 \geq \gamma + 2\kappa$. As a result,

$$
\begin{aligned}
&\sum_{g_{exp}} P(\Psi(G) = g_{exp})P(Y \neq Y'|\Psi(G) = g_{exp}, \quad g_{exp} \subseteq G') \\
&\leq 1 - (1 - \gamma - \kappa)(1 - \gamma - 2\kappa) \\
&= -\gamma^2 - 2\kappa^2 + 2\gamma + 3\kappa - 3\gamma\kappa.
\end{aligned}
$$

$\square$

## A.2 PROOF OF THEOREM 5.4

*Proof.* The proof builds upon the techniques developed for evaluating the sample complexity under transformation invariances in (Shao et al., 2022). However, there are several key differences in the setting under consideration in this work which merits a separate treatment of the problem. First, transformations such as rotation and color translations, considered in prior works on DA in non-graphical domains, form closed groups which facilitate the analysis by focusing on the orbits generated by the group operations. In contrast, the transformations considered in this work perturb the non-explanation subgraph while preserving the explanation subgraph. This does not form a closed group. Second, in prior works, it is assumed that the transformation leads to a similarly labeled samples with probability one, whereas in our setting, the label preservation is probabilistic and depends on the explainability parameters as quantified in Proposition 4.1.

Let us fix $m \in \mathbb{N}$. Let $d = VC_{EA}(\mathcal{H}, \Psi)$. Consider two sets $\mathcal{T}$ and $\mathcal{T}'$ of $m$ independently generated graph and label pairs generated according to $P_{G,Y}$. We first note that:

$$P(err_{P_{G,Y}}(\widetilde{f}) \geq err_{P_{G,Y}}(f^*) + 2\epsilon)$$

$$\leq P(err_{P_{G,Y}}(\widetilde{f}) \geq err_{\mathcal{T}}(f) + \epsilon \text{ or } err_{\mathcal{T}}(f) > err_{\mathcal{T}}(f^*) \text{ or } err_{\mathcal{T}}(f^*) \geq err_{P_{G,Y}}(f^*) + \epsilon)$$

$$\leq P(err_{P_{G,Y}}(\widetilde{f}) \geq err_{\mathcal{T}}(f) + \epsilon)$$

$$+ P(err_{\mathcal{T}}(f) > err_{\mathcal{T}}(f^*))$$

$$+ P(err_{\mathcal{T}}(f^*) \geq err_{P_{G,Y}}(f^*) + \epsilon)$$

where $err_{\mathcal{T}}(\cdot)$ denotes the empirical error over the set $\mathcal{T}$, $err_{P_{G,Y}}$ denotes the statistical error with respect to $P_{G,Y}$, $f(\cdot)$ and $\widetilde{f}(\cdot)$ are defined as in equation 2, and $f^*$ is the optimal classifier in the hypothesis class in terms of statistical error rate. We bound each of the three terms separately. We first bound

$$P(err_{P_{G,Y}}(\widetilde{f}) \geq err_{\mathcal{T}}(f) + \epsilon)$$

Let us denote the event

$$\mathcal{E}_{\mathcal{T},\epsilon} \triangleq \{\exists f \in \mathcal{H} : err_{P_{G,Y}}(\widetilde{f}) \geq err_{\mathcal{T}}(f) + \epsilon\}.$$

We provide sufficient conditions on $m$ under which $P(\mathcal{E}_{\mathcal{T},\epsilon}) \leq \frac{\delta}{2}$. To this end, let us define

$$\mathcal{E}_{\mathcal{T},\mathcal{T}',\epsilon} = \{\exists f \in \mathcal{H} : err_{\mathcal{T}'}(\widetilde{f}) \geq err_{\mathcal{T}}(f) + \frac{\epsilon}{2}\}.$$

Note that:

$$P(\mathcal{E}_{\mathcal{T},\mathcal{T}',\epsilon}) \geq P(\mathcal{E}_{\mathcal{T},\epsilon}, \mathcal{E}_{\mathcal{T},\mathcal{T}',\epsilon}) = P(\mathcal{E}_{\mathcal{T},\epsilon})P(\mathcal{E}_{\mathcal{T},\mathcal{T}',\epsilon}|\mathcal{E}_{\mathcal{T},\epsilon}), \quad (6)$$

Consequently, to derive an upper-bound on $P(\mathcal{E}_{\mathcal{T},\epsilon})$ it suffices to derive a lower-bound on $P(\mathcal{E}_{\mathcal{T},\mathcal{T}',\epsilon}|\mathcal{E}_{\mathcal{T},\epsilon})$ and an upper-bound on $P(\mathcal{E}_{\mathcal{T},\mathcal{T}',\epsilon})$. We first derive a lower-bound on $P(\mathcal{E}_{\mathcal{T},\mathcal{T}',\epsilon}|\mathcal{E}_{\mathcal{T},\epsilon})$. Note that:

$$P(\mathcal{E}_{\mathcal{T},\mathcal{T}',\epsilon}|\mathcal{E}_{\mathcal{T},\epsilon}) = P(\exists f' \in \mathcal{H} : err_{\mathcal{T}'}(\widetilde{f'}) \geq err_{\mathcal{T}}(f') + \frac{\epsilon}{2}|\exists f \in \mathcal{H} : err_{P_{G,Y}}(\widetilde{f}) \geq err_{\mathcal{T}}(f) + \epsilon)$$

$$\geq P(err_{\mathcal{T}'}(\widetilde{f}) \geq err_{\mathcal{T}}(f) + \frac{\epsilon}{2}|err_{P_{G,Y}}(\widetilde{f}) \geq err_{\mathcal{T}}(f) + \epsilon)$$

$$= \sum_{(g_i,y_i),i\in[m]} P(\mathcal{T} = \{(g_i,y_i)|i \in [m]\}) \times$$

$$P(err_{\mathcal{T}'}(\widetilde{f}) \geq err_{\mathcal{T}}(f) + \frac{\epsilon}{2}|err_{P_{G,Y}}(\widetilde{f}) \geq err_{\mathcal{T}}(f) + \epsilon, \mathcal{T} = \{(g_i,y_i)|i \in [m]\})$$

For a given realization of the training set $\mathcal{T} = \{(g_i,y_i), i \in [m]\}$, let $err_{\mathcal{T}}(f) + \epsilon$ be denoted by the (constant) variable $c_{\mathcal{T}}$. Then, we have:

$$P(\mathcal{E}_{\mathcal{T},\mathcal{T}',\epsilon}|\mathcal{E}_{\mathcal{T},\epsilon}) = \sum_{(\overline{g}_i,y_i),i\in[m]} P(\mathcal{T} = \{(\overline{g}_i,y_i)|i \in [m]\}) \times$$

$$P(err_{\mathcal{T}'}(\widetilde{f}) - c_{\mathcal{T}} \geq -\frac{\epsilon}{2}|\mathbb{E}(err_{\mathcal{T}'}(\widetilde{f})) \geq c_{\mathcal{T}})$$

where we have used the fact that $\mathcal{T}'$ is a collection of independent and identically distributed (IID) samples to conclude that $err_{P_{G,Y}}(\widetilde{f}) = \mathbb{E}(err_{\mathcal{T}'}(\widetilde{f}))$. Consequently,

$$P(\mathcal{E}_{\mathcal{T},\mathcal{T}',\epsilon}|\mathcal{E}_{\mathcal{T},\epsilon}) \geq P(err_{\mathcal{T}'}(\widetilde{f}) - \mathbb{E}(err_{\mathcal{T}'}(\widetilde{f})) \geq -\frac{\epsilon}{2}),$$

where we have dropped the condition $\mathbb{E}(err_{\mathcal{T}'}(\widetilde{f})) \geq c_{\mathcal{T}}$ since $err_{\mathcal{T}'}(\widetilde{f}) - \mathbb{E}(err_{\mathcal{T}'}(\widetilde{f}))$ is independent of $\mathcal{T}$, and used the fact that $\sum_{(g_i,y_i),i\in[m]} P(\mathcal{T} = \{(g_i,y_i)|i \in [m]\}) = 1$. Note that $err_{\mathcal{T}'}(\widetilde{f}) = \frac{1}{m}\sum_{(G_i,Y_i)\in\mathcal{T}'} \mathbb{1}(\widetilde{f}(G_i) \neq Y_i)$ and $\mathbb{E}(err_{\mathcal{T}'}(\widetilde{f})) = P(\widetilde{f}(G_i) \neq Y_i), i \in [m]$, where we have used the linearity of expectation. So,

$$P(\mathcal{E}_{\mathcal{T},\mathcal{T}',\epsilon}|\mathcal{E}_{\mathcal{T},,\epsilon}) \geq P(\sum_{(G_i,Y_i)\in\mathcal{T}'} (\mathbb{1}(\widetilde{f}(G_i) \neq Y_i) - P(\widetilde{f}(G_i) \neq Y_i)) \geq \frac{-m\epsilon}{2})$$

$$\geq 1 - e^{-\frac{2m^2\epsilon^2}{m}} = 1 - e^{-2m\epsilon^2},$$

where we have used the Hoeffding's inequality and the fact that $\mathbb{1}(\widetilde{f}(G_i) \neq Y_i) \in [0,1]$. Hence, if $m \geq \frac{2}{\epsilon^2}$, then:

$$P(\mathcal{E}_{\mathcal{T},\mathcal{T}',\epsilon}|\mathcal{E}_{\mathcal{T},\epsilon}) \geq 1 - \frac{1}{e} \geq \frac{1}{2}. \tag{7}$$

Combining equation 6 and equation 7, we get:

$$P(\mathcal{E}_{\mathcal{T},\mathcal{T}',\epsilon}) \geq \frac{1}{2}P(\mathcal{E}_{\mathcal{T},\epsilon}).$$

Hence, to prove $P(\mathcal{E}_{\mathcal{T},\epsilon}) \leq \frac{\delta}{2}$, it suffices to provide sufficient conditions on $m$ such that $P(\mathcal{E}_{\mathcal{T},\mathcal{T}',\epsilon}) \leq \frac{\delta}{4}$.

We first introduce the notion of perturbed subset. For a given set of labeled graphs $\mathcal{U}$ and an element $(G,Y)$ in $\mathcal{U}$, let us define the perturbed set of $G$ in $\mathcal{U}$ as $\mathcal{O}(G|\mathcal{U}) \triangleq \{(G',Y) \in \mathcal{U}|\Psi(G) \in \overline{G}'\}$, i.e., as the set of all elements of $\mathcal{U}$ that can be produced via explanation-preserving edge additions and omissions in $G$. Furthermore, let $n_{\mathcal{O}}(G|\mathcal{U}) \triangleq |\mathcal{O}(G|\mathcal{U})|$ be the number of explanation-preserving perturbations of $G$ in $\mathcal{U}$.

Next, let us define $\mathcal{T}'' = \mathcal{T} \cup \mathcal{T}'$. Let $G_1, G_2, \cdots, G_{2m}$ be the elements of $\mathcal{T}''$ sorted such that $n_{\mathcal{O}}(G_i|\mathcal{T}'') \geq n_{\mathcal{O}}(G_j|\mathcal{T}''), j \leq i$, i.e. sorted in a non-decreasing order with respect to $n_{\mathcal{O}}(\cdot|\mathcal{T}'')$ so that there are a larger or equal number of samples which are perturbations of $G_i$ in $\mathcal{T}''$ than that of $G_j$ for all $j \leq i$. We construct the sets $\mathcal{R}_1$ and $\mathcal{R}_2$ partitioning $\mathcal{T}''$ as follows. Initiate $\mathcal{R}_1 = \mathcal{R}_2 = \phi$ and $\mathcal{T}_1'' = \mathcal{T}''$. If $n_{\mathcal{O}}(G_1|\mathcal{T}_1'') > \log^2 m$, we add $\mathcal{O}(G_1|\mathcal{T}_1'')$ to $\mathcal{R}_1$ and construct $\mathcal{T}_2'' = \mathcal{T}'' - \mathcal{O}(G_1|\mathcal{T}'')$. We define $G^{(1)} = G_1$ and call it the subset representative for $\mathcal{O}(G_1|\mathcal{T}'')$. Next, we arrange the elements of $\mathcal{T}_2''$ in a non-decreasing order with respect to $n_{\mathcal{O}}(\cdot|\mathcal{T}_2'')$ similar to the previous step. Let $G^{(2)}$ denote the sample with the largest $n_{\mathcal{O}}(\cdot|\mathcal{T}_1'')$ value. If $n_{\mathcal{O}}(G^{(2)}|\mathcal{T}_2'') \geq \log^2 m$, its corresponding set $\mathcal{O}(G^{(2)}|\mathcal{T}_2'')$ is added to $\mathcal{R}_1$. This process is repeated until the $\ell$th step when $n_{\mathcal{O}}(G^{(\ell)}|\mathcal{T}_\ell)$ is less than $\log^2 m$. Then, we set $\mathcal{R}_2 = \mathcal{T}_\ell$, thus partitioning $\mathcal{T}''$ into two sets. Loosely speaking, $\mathcal{R}_1$ contains the samples which have more than $\log^2 m$ of their explanation-preserving perturbations in non-overlapping subsets of $\mathcal{T}''$, and $\mathcal{R}_2$ contains the samples which, after removing elements of $\mathcal{R}_1$, do not have more that $\log^2 m$ of their perturbations in the other remaining samples.

We further define $\mathcal{S}_i = \mathcal{T} \cap \mathcal{R}_i$ and $\mathcal{S}_i' = \mathcal{T} \cap \mathcal{R}_i', i \in \{1,2\}$. For any collection $\mathcal{A} = \{(g_i, y_i), i \in [|\mathcal{A}|]\}$ of graphs and classification function $f'(\cdot)$, let $\overline{M}_{\mathcal{A}}(f') \triangleq \frac{1}{|\mathcal{A}|}\sum_{(g_i,y_i) \in \mathcal{A}} \mathbb{1}(f'(g_i) \neq y_i)$ be the fraction of missclassifed elements of $\mathcal{A}$ by $f'(\cdot)$. Then,

$$P(\mathcal{E}_{\mathcal{T},\mathcal{T}',\epsilon}) = P(\exists f \in \mathcal{H} : err_{\mathcal{T}'}(\widetilde{f}) \geq err_{\mathcal{T}}(f) + \frac{\epsilon}{2})$$

$$\leq P(\exists f \in \mathcal{H} : \overline{M}_{\mathcal{S}_1'}(\widetilde{f}) \geq \overline{M}_{\mathcal{S}_1}(\widetilde{f}) + \frac{\epsilon}{4} \text{ or } \overline{M}_{\mathcal{S}_2'}(\widetilde{f}) \geq \overline{M}_{\mathcal{S}_2}(\widetilde{f}) + \frac{\epsilon}{4})$$

$$\leq P(\exists f \in \mathcal{H} : \overline{M}_{\mathcal{S}_1'}(\widetilde{f}) \geq \overline{M}_{\mathcal{S}_1}(\widetilde{f}) + \frac{\epsilon}{4}) + P(\exists f \in \mathcal{H} : \overline{M}_{\mathcal{S}_2'}(\widetilde{f}) \geq \overline{M}_{\mathcal{S}_2}(\widetilde{f}) + \frac{\epsilon}{4}). \tag{8}$$

We upper-bound each term in equation 8 separately.

**Step 1:** Finding an upper-bound for the term $P(\exists f \in \mathcal{H} : \overline{M}_{\mathcal{S}_1'}(\widetilde{f}) \geq \overline{M}_{\mathcal{S}_1} + \frac{\epsilon}{4})$:

We first find an upper bound for $\ell$. Note that by construction
i) $\mathcal{R}_1 = \bigcup_{i \in [\ell]} \mathcal{O}(G^{(i)}|\mathcal{T}_i)$,
ii) $n_{\mathcal{O}}(G^{(i)}|\mathcal{T}_i) \geq \log^2 m, i \in [\ell]$,
iii) $\mathcal{O}(G^{(i)}|\mathcal{T}_i)$ are disjoint, and
iv) $|\mathcal{R}_1| \leq |\mathcal{T}''| = 2m$.
From i) and iv), we have $\bigcup_{i \in [\ell]} \mathcal{O}(G^{(i)}|\mathcal{T}_i) \leq 2m$, and from iii), we have $\sum_{i \in [\ell]} |\mathcal{O}(G^{(i)}|\mathcal{T}_i)| = \sum_{i \in \ell} n_{\mathcal{O}}(G^{(i)}|\mathcal{T}_i) \leq 2m$, and from ii), we conclude that $\ell \leq \frac{2m}{\log^2 m}$.

Next, we bound the expected number of missclassified elements of $\mathcal{R}_1$ for which there is at least one training sample in $\mathcal{T}$ with the same explanation subgraph. To this end, let us define:

$$err_{\mathcal{O}} \triangleq \frac{1}{m} \sum_{(G,Y) \in \mathcal{T}''} \mathbb{1}(\widetilde{f}(G) \neq Y \wedge \exists (G',Y') \in \mathcal{T} : \Psi(G') \subseteq G),$$

that is, $err_{\mathcal{O}}$ is the fraction of elements of $\mathcal{T}''$ which are missclassified despite the existence of at least one training sample whose explanation subgraph is a subgraph of the missclassfied graph. Let $\mathcal{G}_{exp}$ be the image of $\Psi(\cdot)$, and for any $g_{exp} \in \mathcal{G}_{exp}$ define

$$err_{\mathcal{O}, g_{exp}} \triangleq \frac{1}{m} \sum_{(G, Y_G) \in \mathcal{T}''} \mathbb{1}(\widetilde{f}(G) \neq Y_G \wedge \exists (G', Y') \in \mathcal{T} : \Psi(G') = g_{exp} \text{and } g_{exp} \subseteq G).$$

Note that $err_{\mathcal{O}} = \sum_{g_{exp} \in \mathcal{G}_{exp}} err_{\mathcal{O}, g_{exp}}$. Furthermore,

$$\mathbb{E}(err_{\mathcal{O}, g_{exp}}) \leq \frac{1}{m} |\mathcal{T}''| P(\Psi(G') = g_{exp}) P(Y_{G'} \neq Y_G | \Psi(G') = g_{exp} \text{ and } g_{exp} \subseteq G).$$

Consequently, from Proposition 4.1, we have

$$\mathbb{E}(err_{\mathcal{O}}) \leq \frac{1}{m} |\mathcal{T}''| \sum_{g_{exp}} P(\Psi(G') = g_{exp}) P(Y_{G'} \neq Y_G | \Psi(G') = g_{exp} \text{ and } g_{exp} \subseteq G) \leq 2\zeta,$$

where $\zeta \triangleq -\gamma^2 - 2\kappa^2 + 2\gamma + 3\kappa - 3\gamma\kappa$.

Consequently, from Hoeffding's inequality, we have $P(err_{\mathcal{O}} \geq 4\zeta) \leq 2^{-m\zeta^2}$. So,

$$P(\exists f \in \mathcal{H} : \overline{M}_{\mathcal{S}_1'}(\widetilde{f}) \geq \overline{M}_{\mathcal{S}_1}(\widetilde{f}) + \frac{\epsilon}{4})$$

$$\leq P(\exists f \in \mathcal{H} : \overline{M}_{\mathcal{S}_1'}(\widetilde{f}) \geq \overline{M}_{\mathcal{S}_1}(\widetilde{f}) + \frac{\epsilon}{4}, err_{\mathcal{O}} \leq 4\zeta) + P(err_{\mathcal{O}} \geq 4\zeta)$$

$$\leq P(\exists f \in \mathcal{H} : \overline{M}_{\mathcal{S}_1'}(\widetilde{f}) \geq \overline{M}_{\mathcal{S}_1}(\widetilde{f}) + \frac{\epsilon}{4}, err_{\mathcal{O}} \leq 4\zeta) + 2^{-m\zeta^2}$$

$$\leq P(\exists f \in \mathcal{H} : \overline{M}_{\mathcal{S}_1'}(\widetilde{f}) \geq \frac{\epsilon}{4}, err_{\mathcal{O}} \leq 4\zeta) + 2^{-m\zeta^2}$$

Let $\mathcal{A}$ be the set of indices of $\mathcal{O}(G^{(i)}|\mathcal{T}_i)$ which contain at least one sample which is missclassified by $\widetilde{f}$. Note that since $err_{\mathcal{O}}(\mathcal{T}) \leq 4\zeta$, at most $4\zeta m$ of the elements in $\cup_{i \in [\ell]} \mathcal{O}(G^{(i)}|\mathcal{T}_i)$ can be in $\mathcal{T}$ and the rest must be in $\mathcal{T}'$. Since $\mathcal{T}$ and $\mathcal{T}'$ are generated identically, each element of $\mathcal{T}''$ is in $\mathcal{T}$ or $\mathcal{T}'$ with equal probability, i.e., with probability equal to $\frac{1}{2}$.

Let $\mathcal{I} \subseteq [\ell]$ be the set of indices of $\mathcal{O}(G^{(i)}|\mathcal{T}_i)$ which have at least one missclassified element. If $|\mathcal{I}| = i$, then $|\cup_{j \in \mathcal{I}} \mathcal{O}(G^{(j)}|\mathcal{T}_j)| \geq \max(i \log^2 m, \frac{m\epsilon}{4})$, by construction. The probability that at most $4\zeta m$ of these elements are in $\mathcal{T}$ is upper bounded by:

$$P(\exists f \in \mathcal{H} : \overline{M}_{\mathcal{S}_1'}(\widetilde{f}) \geq \frac{\epsilon}{4}, err_{\mathcal{O}} \leq 4\zeta)$$

$$= P(|\cup_{j \in \mathcal{I}} \mathcal{O}(G^{(j)}|\mathcal{T}_j) \cap \mathcal{T}| \leq 4\zeta m, |\cup_{j \in \mathcal{I}} \mathcal{O}(G^{(j)}|\mathcal{T}_j) \cap \mathcal{T}'| \geq \frac{m\epsilon}{4})$$

$$\overset{(a)}{\leq} \sum_{i=1}^{\ell} \binom{\frac{2m}{\log^2 m}}{i} \sum_{j=1}^{4\zeta m} \binom{\max(i \log^2 m, \frac{m\epsilon}{4})}{j} 2^{-\max(i \log^2 m, \frac{m\epsilon}{4})}$$

$$\leq \sum_{i=1}^{\ell} \binom{\frac{2m}{\log^2 m}}{i} 4\zeta m \binom{\max(i \log^2 m, \frac{m\epsilon}{4})}{4\zeta m} 2^{-\max(i \log^2 m, \frac{m\epsilon}{4})}$$

$$\leq \sum_{i=1}^{\ell} \binom{\frac{2m}{\log^2 m}}{i} 2^{-\max(i \log^2 m, \frac{m\epsilon}{4}) + 4\zeta m \log \max(i \log^2 m, \frac{m\epsilon}{4}) + \log m}$$

$$= \sum_{i \in [1, \frac{m\epsilon}{4 \log^2 m}]} \binom{\frac{2m}{\log^2 m}}{i} 2^{-\frac{m\epsilon}{4} + 4\zeta m \log \frac{m\epsilon}{4} + \log m}$$

$$+ \sum_{i \in [\frac{m\epsilon}{4 \log^2 m}, \ell]} \binom{\frac{2m}{\log^2 m}}{i} 2^{-i \log^2 m + 4\zeta m \log (i \log^2 m) + \log m}$$

$$\overset{(b)}{\leq} \frac{m\epsilon}{4 \log^2 m} \binom{\frac{2m}{\log^2 m}}{\frac{m\epsilon}{4 \log^2 m}} 2^{-\frac{m\epsilon}{8}} + \ell \max_{i \in [\ell]} \binom{\frac{2m}{\log^2 m}}{i} 2^{-\frac{1}{2} i \log^2 m},$$

where in (a) we have used the union bound and in (b) we have used the fact that $\epsilon \geq 32\zeta$. Consequently,

$$P(\exists f \in \mathcal{H} : \overline{M}_{\mathcal{S}_1'}(\widetilde{f}) \geq \overline{M}_{\mathcal{S}_1}(\widetilde{f}) + \frac{\epsilon}{4}) \leq \frac{m\epsilon}{4\log^2 m} \left( \frac{\frac{2m}{\log^2 m}}{\frac{m\epsilon}{4\log^2 m}} \right) 2^{-\frac{m\epsilon}{8}} + \ell \max_{i \in [\ell]} \left( \frac{\frac{2m}{\log^2 m}}{i} \right) 2^{-\frac{1}{2}i\log^2 m}$$

$$\leq 2^{\frac{-m\epsilon}{16}} + \frac{2m}{\log^2 m} \max_{i \in [\ell]} 2^{\frac{-1}{2}i\log^2 m + i\log 2m} \leq 2^{\frac{-m\epsilon}{16}} + \frac{2m}{\log^2 m} 2^{-\frac{m\epsilon}{32}\log^2 m} \leq 2^{-\frac{m\epsilon}{32}}.$$

So,

$$P(\exists f \in \mathcal{H} : \overline{M}_{\mathcal{S}_1'}(\widetilde{f}) \geq \frac{\epsilon}{4}, err_{\mathcal{O}} \leq 4\zeta) \leq 2^{\frac{-\epsilon m}{32}} + 2^{-m\zeta^2} \leq 2 \cdot 2^{\frac{-\epsilon m}{32}},$$

where we have used the fact that $1 \geq \epsilon \geq 32\zeta$.

**Step 2:** Finding an upper-bound for the term $P(\exists f \in \mathcal{H} : \overline{M}_{\mathcal{S}_2'}(\widetilde{f}) \geq \overline{M}_{\mathcal{S}_2}(\widetilde{f}) + \frac{\epsilon}{4})$:

By definition of $VC_{EA}(\mathcal{H}, \Psi)$, the number of points in $\mathcal{R}_2$ which can be shattered by $\mathcal{H}$ is at most $d\log^2 m$, where $d \triangleq VC_{EA}(\mathcal{H}, \Psi)$. Let $\mathcal{K}$ be the set of all possible ways to labeling $\mathcal{T}''$ by $\mathcal{H}$. Then, $|\mathcal{K}| \leq \sum_{i=0}^{d\log^2(m)} \binom{2m}{i} \leq (\frac{2em}{d})^{d\log^2(m)}$ by Sauer's lemma. On the other hand:

$$P(\exists f \in \mathcal{H} : \overline{M}_{\mathcal{S}_2'}(\widetilde{f}) \geq \overline{M}_{\mathcal{S}_2}(\widetilde{f}) + \frac{\epsilon}{4})$$

$$\leq \sum_{K \in \mathcal{K}} P(\overline{M}_{\mathcal{S}_2'} \geq \mathbb{E}(\overline{M}_{\mathcal{S}_2'}) + \frac{\epsilon}{8} \text{ or } \overline{M}_{\mathcal{S}_2} \leq \mathbb{E}(\overline{M}_{\mathcal{S}_2}) - \frac{\epsilon}{8}|K)$$

$$\leq (\frac{2em}{d})^{d\log^2(m)}(P(\overline{M}_{\mathcal{S}_2'} \geq \mathbb{E}(\overline{M}_{\mathcal{S}_2'}) + \frac{\epsilon}{8}|K) + P(\overline{M}_{\mathcal{S}_2} \leq \mathbb{E}(\overline{M}_{\mathcal{S}_2}) - \frac{\epsilon}{8}|K))$$

$$\overset{(a)}{\leq} 2(\frac{2em}{d})^{d\log^2(m)} e^{-2m(\frac{\epsilon}{8}^2)} \leq e^{-\frac{m\epsilon^2}{32} + d\log^2(m)\ln\frac{2em}{d}},$$

where we have used Hoeffding's inequality in (a). Taking $m > \frac{32}{\epsilon^2}\left(d\log^2(m)ln(\frac{2em}{d}) + ln(\frac{8}{\delta})\right) + \frac{32}{\epsilon}log(\frac{8}{\delta})$, we get $P(\mathcal{E}_{\mathcal{T},\frac{\epsilon}{2}}) \leq 2P(\mathcal{E}_{\mathcal{T},\mathcal{T}',\frac{1}{2}\epsilon}) \leq 2(\frac{\delta}{8} + \frac{\delta}{8}) = \frac{\delta}{2}$. Then, as described at the beginning of the proof,

$$P(err_{P_G}(\widetilde{f}) \geq err_{P_G}(f^*) + \epsilon) \tag{9}$$

$$\leq P(err_{P_G}(\widetilde{f}) \geq err_{\mathcal{T}}(f) + \frac{1}{2}\epsilon \text{ or } err_{\mathcal{T}}(f) > err_{\mathcal{T}}(f^*) \text{ or } err_{\mathcal{T}}(f^*) \geq err_{P_G}(f^*) + \frac{1}{2}\epsilon) \tag{10}$$

$$\leq P(err_{P_G}(\widetilde{f}) \geq err_{\mathcal{T}}(f) + \frac{1}{2}\epsilon) + P(err_{\mathcal{T}}(f) > err_{\mathcal{T}}(f^*)) + P(err_{\mathcal{T}}(f^*) \geq err_{P_G}(f^*) + \frac{1}{2}\epsilon) \tag{11}$$

$$\leq P(\mathcal{E}_{\mathcal{T},\frac{\epsilon}{2}}) + 0 + \frac{\delta}{2} \leq \frac{\delta}{2} + \frac{\delta}{2} = \delta, \tag{12}$$

where in equation 11 we have used the union bound, and in equation 12 we have used Hoeffding's inequality to conclude that $P(err_{\mathcal{T}}(f^*) \geq err_{P_G}(f^*) + \frac{1}{2}\epsilon) \leq \frac{\delta}{2}$ and the definition of EA-ERM to conclude that $P(err_{\mathcal{T}}(f) > err_{\mathcal{T}}(f^*)) = 0$. Consequently,

$$m_{EA}(\epsilon, \delta, \kappa, \gamma; \mathcal{H}, \Psi) = O\left(\frac{d}{\epsilon^2}\log^2 d + \frac{1}{\epsilon^2}ln(\frac{1}{\delta})\right)$$

$\square$

# B  DETAILED EXPERIMENTAL SETUP

Our experiments were conducted on a Linux system equipped with eight NVIDIA A100 GPUs, each possessing 40GB of memory. We use CUDA version 11.3, Python version 3.7.16, and Pytorch version 1.12.1.

## B.1 DATASETS

In our empirical experiments, we use a benchmark synthetic dataset and 6 real-life datasets.

- **BA-2motifs** (Luo et al., 2020) dataset includes 1,000 synthetic graphs created from the basic Barabasi-Albert (BA) model. This dataset is divided into two different categories: half of the graphs are associated with 'house 'motifs, while the other half are integrated with five-node circular motifs. The labels of these graphs depends on the specific motif they incorporate.

- **MUTAG** (Debnath et al., 1991) dataset comprises 2,951 molecular graphs, divided into two classes according to their mutagenic effects on the Gram-negative bacterium S. Typhimurium. Functional groups $NO_2$ and $NH_2$ are considered as ground truth explanations for positive samples (Luo et al., 2020).

- **Benzene** (Sánchez-Lengeling et al., 2020) is a dataset of 12,000 molecular graphs from the ZINC15 database(Sterling & Irwin, 2015). The graphs are divided into two classes based on whether they have a benzene ring or not. If a molecule has more than one benzene ring, each ring is a separate explanation.

- **Fluoride** (Sánchez-Lengeling et al., 2020) dataset contains 8,671 molecular graphs, divided into two classes based on whether they have both a fluoride and a carbonyl group or not. The ground truth explanations are based on the specific combinations of fluoride atoms and carbonyl functional groups found in each molecule.

- **Alkane** (Sánchez-Lengeling et al., 2020) is a dataset of 4,326 molecular graphs, divided into two classes. A positive sample is a molecule with an unbranched alkane and a carbonyl group.

- **D&D** (Dobson & Doig, 2003) comprises 1,178 protein structures. proteins are depicted as graphs where each node represents an amino acid. Nodes are interconnected by an edge if the amino acids are within 6 Angstroms of each other. Protein structures into binary classes: enzymes and non-enzymes.

- **PROTEINS** (Dobson & Doig, 2003; Borgwardt et al., 2005) consists of 1,113 protein graphs which are generated in the same way as D&D.

The statistics of datasets are shown in Table 2. The # of explanations denotes the number of graphs with ground truth explanations.

Table 2: The detailed information of graph datasets

| Dataset | #graphs | #nodes | #edges | #explanations | #classes |
|---|---|---|---|---|---|
| BA-2motifs | 1,000 | 25 | 50-52 | 1,000 | 2 |
| MUTAG | 2,951 | 5-417 | 8-224 | 1,015 | 2 |
| Benzene | 12,000 | 4-25 | 6-58 | 6,001 | 2 |
| Fluoride | 8,671 | 5-25 | 8-58 | 1,527 | 2 |
| Alkane | 4,326 | 5-25 | 8-58 | 375 | 2 |
| D&D | 1,178 | 30-5,748 | 126 -28,534 | 0 | 2 |
| PROTEINS | 1,113 | 4-620 | 10-2,098 | 0 | 2 |

## B.2 GNN MODELS

We use the same GCN model architectures and hyperparameters as (Luo et al., 2020). Specifically, For the GCN model, we embed the nodes with two GCN-Relu-BatchNorm blocks and one GCN-Relu block to learn node embeddings. Then, we adopt readout operations (Xu et al., 2019) to get graph embeddings, followed by a linear layer for graph classification. The number of neurons is set to 20 for hidden layers. For the GIN model, we replace the GCN layer with a Linear-Relu-Linear-Relu GIN layer. For the PNA model, we adopt a similar architecture in (Miao et al., 2022). We initialize the variables with the Pytorch default setting and train the models with Adam optimizer with a learning rate of $1.0 \times 10^{-3}$.

### B.3 DATA AUGMENTATION BASELINES

- Edge Inserting: We randomly select 10% unconnected node pairs to generate the augmentation graph.

- Edge Dropping: We generate a graph by randomly removing 10% edges in the original graph.

- Node Dropping: We generate a graph by randomly dropping 10% nodes from the input graph, together with their associated edges

- Feature Dropping: We generate a graph by randomly dropping 10% features.

- Mixup: Given a labeled graph $(G_i, Y_i)$, we randomly sample another labelled graph $(G_j, Y_j)$. There adjacency matrices are denoted by $\boldsymbol{A}_i$ and $\boldsymbol{A}_j$, respectively. We generate a block diagonal matrix $\text{diag}(\boldsymbol{A}_i, \boldsymbol{A}_j)$:

$$\text{diag}(\boldsymbol{A}_i, \boldsymbol{A}_j) = \left[ \begin{array}{cc} \boldsymbol{A}_i & 0 \\ 0 & \boldsymbol{A}_j \end{array} \right]. \tag{13}$$

The corresponding graph is denoted as $G^{(\text{diag})}$ We obtain the mixup augmentation graph by randomly adding two cross-graph edges, i.e., one node from $G_i$ and the other from $G_j$, to $G^{(\text{diag})}$.

## C EXTRA EXPERIMENTS

In this section, we provide extensive experiments to further verify the effectiveness of our method and support our theoretical findings.

### C.1 ANALYSIS ON DISTRIBUTION OF OUR METHOD

As described in Section 7, we generate augmentations by an in-distributed perturbation function $\Pi(\cdot)$ (Algorithm 2). In this part, we empirically verify the effectiveness of our implementation in generating in-distributed augmentations. We use both GNNExplainer (Ying et al., 2019) and PGExplainer (Luo et al., 2020) to generate explanations. Two real-life datasets, Fluoride and Alkane are utilized here. For each dataset, we first pad each graph by inserting isolated nodes such that all graphs have the same size of nodes. Then, for each graph, we concatenate its adjacency matrix with the node matrix followed by a flatten operation to get a high-dimensional vector. We adopt an encoder network to embed high-dimensional vectors into a 2-D vector space. The encoder network consists of two fully connected layers, the same as the decoder network. Cross Entropy is used as the reconstruction error to train the Autoencoder model. The original graphs and augmentations are used for training. The visualization results of these original and augmentation graphs are shown in Figure 3. We observe that augmentation graphs are in-distributed in both datasets.

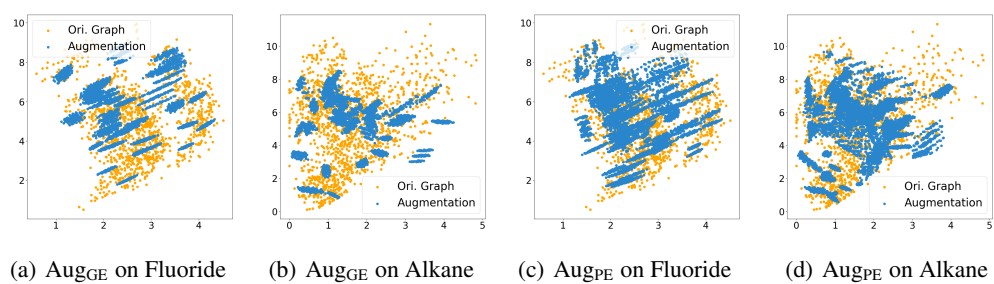

| (a) Aug$_{\text{GE}}$ on Fluoride | (b) Aug$_{\text{GE}}$ on Alkane | (c) Aug$_{\text{PE}}$ on Fluoride | (d) Aug$_{\text{PE}}$ on Alkane |

Figure 3: Visualization results of augmentations generated by Aug$_{\text{PE}}$ and Aug$_{\text{PE}}$ (best viewed in color).

### C.2 DEALING WITH OOD AUGMENTATIONS

In this section, we conduct experiments to verify the effectiveness of our strategy that includes a hyperparameter $\lambda$ in alleviating the negative effects of OOD graph augmentations. We select 500,

100, and 100 samples in BA-2motifs dataset as the training set, valid set, and test set, respectively. To obtain OOD augmentations, we add edges to the non-explanation subgraphs until the average node degree is not less than 17. Each training instance has 2 augmentations, and we use 3 layers GCN as the backbone. As Figure C.2 shows, in general, the accuracy decreases as the hyperparameter $\lambda$ rises. The results show that with out-of-distribution graph augmentations, a small $\lambda$ can alleviate the negative effects.

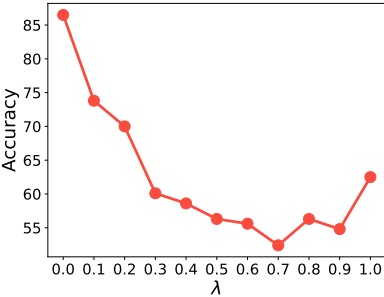

Figure 4: The effects of $\lambda$ in tackling OOD augmentations on BA-2motifs dataset.

### C.3 HYPER-PARAMETER SENSITIVITY STUDIES

In this section, we show the robustness of our method with a set of hyper-parameter sensitivity studies. Two real datasets, MUTAG and Benzene, are used in this part. We choose PGExplainer to generate explanations.

As shown in Algorithm 1, $M$ denotes the number of augmentation samples per instance. We range the values of $M$ from 1 to 30 and show the accuracy performances of GNN models in Figure 5. Our method is robust to the selection of $M$.

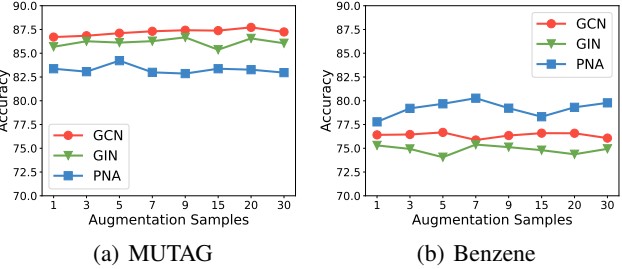

(a) MUTAG          (b) Benzene

Figure 5: Hyper-parametey study of sampling number $M$ by using PGExplainer.

### C.4 COMPARISON TO BASELINE DATA AUGMENTATIONS WITH 3 LAYER PNA.

In this part, we provide the comparison of our methods to baseline data augmentations with the 3-layer PNA and GraphSage as the classifier. As shown in Table 3. we have similar observations with results on GCN and GIN. Our method consistently outperforms other baselines. Specifically, the improvements of $Aug_{GE}$ and $Aug_{PE}$ over the best of others are 1.99% and 1.01% on PNA and 4.91% and 5.35% on GraphSage, respectively.

### C.5 COMPARISON TO BASELINE DATA AUGMENTATIONS WITH SMALLER TRAINING SIZES

We show the performances of GNNs with smaller training sizes to further verify the effectiveness of our methods in improving data efficiency. We consider training sizes with 10 samples and 30 samples in this part. We use GIN in this part and keep other settings the same as Section 8.1. We also include the default setting with 50 samples for comparison.

Table 3: Performance comparisons with 3-layer PNA and GraphSage trained on 50 samples. The metric is classification accuracy. The best results are shown in bold font and the second best ones are underlined.

| | Dataset | MUTAG | Benzene | Fluoride | Alkane | D&D | PROTEINS |
|---|---|---|---|---|---|---|---|
| PNA | Vanilla | $83.61_{\pm 4.13}$ | $76.72_{\pm 3.28}$ | $64.90_{\pm 6.88}$ | $93.60_{\pm 1.72}$ | $61.29_{\pm 5.38}$ | $64.91_{\pm 5.62}$ |
| | Edge Inserting | $82.35_{\pm 3.31}$ | $78.70_{\pm 2.29}$ | $64.17_{\pm 3.76}$ | $89.66_{\pm 4.55}$ | $61.81_{\pm 4.35}$ | $61.91_{\pm 4.44}$ |
| | Edge Dropping | $82.18_{\pm 2.68}$ | $77.72_{\pm 1.14}$ | $62.78_{\pm 1.76}$ | $92.31_{\pm 2.94}$ | $61.64_{\pm 6.22}$ | $60.45_{\pm 4.35}$ |
| | Node Dropping | $81.60_{\pm 2.39}$ | $78.02_{\pm 1.46}$ | $61.59_{\pm 3.15}$ | $90.77_{\pm 3.54}$ | $62.67_{\pm 4.28}$ | $61.64_{\pm 5.30}$ |
| | Feature Dropping | $81.46_{\pm 3.44}$ | $79.01_{\pm 1.74}$ | $66.63_{\pm 3.62}$ | $90.94_{\pm 2.94}$ | $62.59_{\pm 4.53}$ | $61.18_{\pm 3.35}$ |
| | Mixup | $72.21_{\pm 4.20}$ | $75.05_{\pm 0.93}$ | $61.10_{\pm 1.18}$ | $89.31_{\pm 2.37}$ | $57.45_{\pm 2.07}$ | $53.71_{\pm 3.36}$ |
| | Aug$_{GE}$ | $\mathbf{84.73}_{\pm \mathbf{2.04}}$ | $\mathbf{80.61}_{\pm \mathbf{0.65}}$ | $\mathbf{68.72}_{\pm \mathbf{1.55}}$ | $\mathbf{94.97}_{\pm \mathbf{1.46}}$ | $\mathbf{65.43}_{\pm \mathbf{6.44}}$ | $64.64_{\pm 5.21}$ |
| | Aug$_{PE}$ | $\underline{84.15}_{\pm 2.25}$ | $\underline{80.47}_{\pm 0.82}$ | $\underline{68.60}_{\pm 1.96}$ | $93.14_{\pm 2.52}$ | $\underline{63.19}_{\pm 5.23}$ | $\mathbf{65.09}_{\pm \mathbf{4.62}}$ |
| GraphSage | Vanilla | $\underline{88.03}_{\pm 2.18}$ | $68.56_{\pm 6.99}$ | $63.96_{\pm 3.10}$ | $93.40_{\pm 4.16}$ | $64.83_{\pm 3.75}$ | $66.82_{\pm 4.09}$ |
| | Edge Inserting | $86.70_{\pm 2.68}$ | $76.06_{\pm 3.34}$ | $62.09_{\pm 2.36}$ | $94.29_{\pm 3.06}$ | $61.90_{\pm 4.81}$ | $66.18_{\pm 5.62}$ |
| | Edge Dropping | $87.00_{\pm 1.58}$ | $74.50_{\pm 3.16}$ | $62.52_{\pm 2.34}$ | $93.80_{\pm 3.41}$ | $64.83_{\pm 4.67}$ | $67.36_{\pm 5.55}$ |
| | Node Dropping | $86.87_{\pm 1.98}$ | $75.00_{\pm 3.27}$ | $62.26_{\pm 2.43}$ | $93.46_{\pm 3.47}$ | $66.03_{\pm 5.48}$ | $68.36_{\pm 3.94}$ |
| | Feature Dropping | $87.38_{\pm 1.56}$ | $74.76_{\pm 3.20}$ | $63.12_{\pm 3.08}$ | $93.60_{\pm 3.26}$ | $63.36_{\pm 5.89}$ | $66.27_{\pm 5.60}$ |
| | Mixup | $78.71_{\pm 1.77}$ | $52.97_{\pm 2.12}$ | $54.48_{\pm 1.26}$ | $75.23_{\pm 3.38}$ | $62.50_{\pm 4.25}$ | $60.64_{\pm 1.68}$ |
| | Aug$_{GE}$ | $\mathbf{88.47}_{\pm \mathbf{1.15}}$ | $\mathbf{78.69}_{\pm \mathbf{2.88}}$ | $\underline{67.10}_{\pm 1.07}$ | $\mathbf{95.03}_{\pm \mathbf{1.02}}$ | $\underline{67.84}_{\pm 4.41}$ | $\underline{70.27}_{\pm 5.49}$ |
| | Aug$_{PE}$ | $\mathbf{88.47}_{\pm \mathbf{1.52}}$ | $\underline{78.30}_{\pm 2.64}$ | $\mathbf{67.38}_{\pm \mathbf{1.57}}$ | $\underline{94.91}_{\pm 1.00}$ | $\mathbf{68.53}_{\pm \mathbf{3.58}}$ | $\mathbf{70.82}_{\pm \mathbf{4.71}}$ |

From Table 4, we observe that Aug$_{GE}$ and Aug$_{PE}$ improves the accuracy performances by similar margins with smaller training sizes. Specifically, the improvements are 4.38% and 4.45% with 10 training samples, and 5.75% and 5.83% training samples. The results are consistent with Section 8.2, which further verify the effectiveness of our method in boosting the data efficiency for GNN training.

## C.6 COMPARISON TO BASELINE DATA AUGMENTATIONS WITH 1 LAYER GNNS

In this set of experiments, we analyze the effectiveness of our methods on less powerful GNNs. We reduce the GNN layers to 1 for GCN, GIN, and PNA. Other settings are kept the same as in Section 8.1. As the results are shown in Table 5, our methods with GNNExplainer and PGExplainer occupy the best and second-best positions than other six baselines, respectively. Specifically, our methods achieve 3.69%, 3.99%, 4.04% improvements with GNNExplainer and 3.89%, 3.65%, 3.27% improvements with PGExplainer on average with GCN, GIN, and PNA backbones. Similar to the results of Section 8.1, these results show that our methods can enhance the GNN performance on both powerful and less powerful GNNs.

## C.7 COMPARISON TO INVARIANT METHODS WITH 3 LAYER GCNS

We compare our method with invariant methods including GSAT(Miao et al., 2022), DIR(Wu et al., 2022), and GALA(Chen et al., 2024). As the results show in Table 6, our method achieves the best results in most cases. Notably, our method is a kind of data augmentation operation by using the explanation subgraphs as domain invariant variables rather than capturing the invariant subgraphs and optimizing the parameters. From Table 6, our method achieves better results than vanilla consistently but invariant methods achieve worse results than vanilla in most cases.

Table 4: Performance comparisons with 3-layer GIN trained on 10/30/50(default) samples. The metric is classification accuracy. The best results are shown in bold font and the second best ones are underlined.

| | Dataset | MUTAG | Benzene | Fluoride | Alkane | D&D | PROTEINS |
|---|---|---|---|---|---|---|---|
| **10 training samples** | Vanilla | $80.14_{\pm5.03}$ | $60.42_{\pm6.10}$ | $61.62_{\pm2.65}$ | $74.78_{\pm9.24}$ | $60.18_{\pm5.89}$ | $60.52_{\pm9.50}$ |
| | Edge Inserting | $79.05_{\pm3.85}$ | $64.37_{\pm4.44}$ | $60.07_{\pm4.84}$ | $67.89_{\pm12.85}$ | $55.45_{\pm6.62}$ | $58.62_{\pm4.06}$ |
| | Edge Dropping | $74.80_{\pm3.97}$ | $65.13_{\pm2.64}$ | $60.20_{\pm4.10}$ | $74.65_{\pm10.97}$ | $59.27_{\pm4.65}$ | $59.05_{\pm5.96}$ |
| | Node Dropping | $76.90_{\pm4.74}$ | $64.88_{\pm2.63}$ | $59.87_{\pm4.45}$ | $80.14_{\pm10.98}$ | $58.27_{\pm5.07}$ | $58.71_{\pm7.95}$ |
| | Feature Dropping | $75.51_{\pm3.98}$ | $60.95_{\pm5.27}$ | $61.27_{\pm5.50}$ | $67.08_{\pm12.35}$ | $56.64_{\pm5.16}$ | $58.97_{\pm6.30}$ |
| | Mixup | $74.35_{\pm2.09}$ | $51.29_{\pm1.36}$ | $52.25_{\pm2.12}$ | $62.76_{\pm2.51}$ | $62.27_{\pm3.60}$ | $\mathbf{64.66_{\pm3.74}}$ |
| | Aug$_{GE}$ | $\underline{83.47_{\pm2.89}}$ | $\mathbf{69.55_{\pm0.91}}$ | $\mathbf{66.54_{\pm1.96}}$ | $\underline{83.24_{\pm5.82}}$ | $\mathbf{65.45_{\pm3.47}}$ | $63.62_{\pm5.48}$ |
| | Aug$_{PE}$ | $\mathbf{87.48_{\pm2.26}}$ | $\mathbf{69.55_{\pm0.91}}$ | $\underline{63.00_{\pm3.49}}$ | $\mathbf{84.54_{\pm6.14}}$ | $\underline{64.91_{\pm2.34}}$ | $\underline{63.88_{\pm3.07}}$ |
| **30 training samples** | Vanilla | $81.09_{\pm3.58}$ | $64.37_{\pm6.14}$ | $66.03_{\pm2.61}$ | $81.41_{\pm12.50}$ | $62.36_{\pm4.85}$ | $65.00_{\pm7.29}$ |
| | Edge Inserting | $82.07_{\pm3.87}$ | $68.92_{\pm3.07}$ | $64.92_{\pm3.60}$ | $86.08_{\pm8.59}$ | $61.91_{\pm5.63}$ | $64.40_{\pm4.03}$ |
| | Edge Dropping | $80.41_{\pm4.02}$ | $67.48_{\pm4.05}$ | $61.12_{\pm4.10}$ | $88.00_{\pm7.09}$ | $64.09_{\pm4.01}$ | $62.41_{\pm5.82}$ |
| | Node Dropping | $81.02_{\pm4.90}$ | $67.93_{\pm3.50}$ | $60.57_{\pm4.66}$ | $87.61_{\pm7.39}$ | $64.36_{\pm3.14}$ | $64.22_{\pm3.32}$ |
| | Feature Dropping | $81.60_{\pm4.25}$ | $63.88_{\pm5.67}$ | $64.85_{\pm6.23}$ | $85.28_{\pm6.12}$ | $62.09_{\pm5.14}$ | $63.79_{\pm3.99}$ |
| | Mixup | $72.24_{\pm2.55}$ | $54.28_{\pm2.06}$ | $52.06_{\pm3.14}$ | $68.31_{\pm3.98}$ | $56.00_{\pm2.04}$ | $60.95_{\pm3.32}$ |
| | Aug$_{GE}$ | $\mathbf{84.90_{\pm1.29}}$ | $\underline{70.69_{\pm1.66}}$ | $\mathbf{71.56_{\pm4.59}}$ | $\mathbf{94.50_{\pm1.32}}$ | $\mathbf{68.55_{\pm5.64}}$ | $\underline{69.05_{\pm4.23}}$ |
| | Aug$_{PE}$ | $\underline{84.73_{\pm1.45}}$ | $\mathbf{70.81_{\pm2.30}}$ | $\underline{71.48_{\pm3.64}}$ | $\underline{94.28_{\pm1.42}}$ | $\underline{68.00_{\pm6.26}}$ | $\mathbf{70.17_{\pm3.50}}$ |
| **50 training samples** | Vanilla | $82.52_{\pm3.71}$ | $67.48_{\pm5.93}$ | $68.55_{\pm5.18}$ | $85.06_{\pm10.27}$ | $65.14_{\pm4.26}$ | $66.45_{\pm4.01}$ |
| | Edge Inserting | $82.79_{\pm3.21}$ | $71.58_{\pm2.77}$ | $66.78_{\pm4.04}$ | $87.54_{\pm10.32}$ | $64.74_{\pm5.38}$ | $65.45_{\pm5.82}$ |
| | Edge Dropping | $81.63_{\pm3.65}$ | $70.46_{\pm4.34}$ | $62.91_{\pm5.06}$ | $90.29_{\pm6.39}$ | $66.72_{\pm3.76}$ | $62.73_{\pm5.29}$ |
| | Node Dropping | $82.18_{\pm3.99}$ | $71.31_{\pm2.71}$ | $64.86_{\pm4.55}$ | $88.89_{\pm7.00}$ | $66.72_{\pm2.89}$ | $65.64_{\pm5.38}$ |
| | Feature Dropping | $82.72_{\pm2.92}$ | $70.66_{\pm2.80}$ | $67.58_{\pm5.12}$ | $83.09_{\pm11.73}$ | $68.19_{\pm4.34}$ | $65.55_{\pm5.00}$ |
| | Mixup | $74.52_{\pm1.61}$ | $59.00_{\pm3.43}$ | $51.58_{\pm2.59}$ | $65.80_{\pm4.13}$ | $58.55_{\pm3.48}$ | $62.16_{\pm2.92}$ |
| | Aug$_{GE}$ | $\underline{85.99_{\pm2.41}}$ | $\mathbf{75.41_{\pm0.82}}$ | $\underline{76.29_{\pm2.05}}$ | $\mathbf{94.89_{\pm1.11}}$ | $\mathbf{69.31_{\pm5.19}}$ | $\mathbf{68.45_{\pm5.86}}$ |
| | Aug$_{PE}$ | $\mathbf{86.87_{\pm1.79}}$ | $\underline{75.39_{\pm1.03}}$ | $\mathbf{76.49_{\pm1.68}}$ | $\underline{94.77_{\pm1.14}}$ | $\underline{67.41_{\pm2.75}}$ | $\underline{68.09_{\pm5.52}}$ |

Table 5: Accuracy performance comparison using 1 layer GNNs among datasets with 50 samples. We highlight the best and second performance by bold and underlining.

| | Dataset | MUTAG | Benzene | Fluoride | Alkane | D&D | PROTEINS |
|---|---|---|---|---|---|---|---|
| GCN | Vanilla | $79.83_{\pm 3.21}$ | $61.46_{\pm 4.39}$ | $56.77_{\pm 4.46}$ | $94.89_{\pm 1.65}$ | $61.38_{\pm 6.70}$ | $66.45_{\pm 6.95}$ |
| | Edge Inserting | $79.42_{\pm 3.95}$ | $65.36_{\pm 4.91}$ | $54.84_{\pm 3.55}$ | $92.71_{\pm 3.79}$ | $65.26_{\pm 7.58}$ | $62.18_{\pm 4.69}$ |
| | Edge Dropping | $78.10_{\pm 4.50}$ | $66.02_{\pm 4.13}$ | $54.92_{\pm 3.47}$ | $94.40_{\pm 1.76}$ | $66.12_{\pm 7.20}$ | $62.18_{\pm 4.57}$ |
| | Node Dropping | $78.54_{\pm 4.24}$ | $66.19_{\pm 4.04}$ | $54.49_{\pm 3.43}$ | $93.54_{\pm 2.96}$ | $66.21_{\pm 6.11}$ | $63.00_{\pm 5.65}$ |
| | Feature Dropping | $79.56_{\pm 3.80}$ | $63.92_{\pm 4.68}$ | $54.46_{\pm 3.55}$ | $93.11_{\pm 2.79}$ | $66.55_{\pm 3.93}$ | $62.91_{\pm 4.30}$ |
| | Mixup | $56.67_{\pm 2.11}$ | $50.11_{\pm 0.42}$ | $51.94_{\pm 0.53}$ | $61.71_{\pm 0.00}$ | $59.09_{\pm 0.00}$ | $56.12_{\pm 0.46}$ |
| | $\text{Aug}_{GE}$ | $\underline{84.63}_{\pm 1.65}$ | $\underline{69.63}_{\pm 2.03}$ | $\mathbf{61.81}_{\pm \mathbf{1.54}}$ | $\underline{95.69}_{\pm 1.19}$ | $\underline{66.98}_{\pm 5.38}$ | $\underline{66.82}_{\pm 4.98}$ |
| | $\text{Aug}_{PE}$ | $\mathbf{85.17}_{\pm \mathbf{1.63}}$ | $\mathbf{69.64}_{\pm \mathbf{2.07}}$ | $\underline{61.34}_{\pm 1.41}$ | $\mathbf{95.86}_{\pm \mathbf{1.14}}$ | $\mathbf{67.67}_{\pm \mathbf{5.82}}$ | $\mathbf{66.91}_{\pm \mathbf{4.83}}$ |
| GIN | Vanilla | $81.39_{\pm 1.71}$ | $63.71_{\pm 3.33}$ | $60.31_{\pm 4.52}$ | $87.60_{\pm 5.40}$ | $66.21_{\pm 8.48}$ | $66.45_{\pm 3.85}$ |
| | Edge Inserting | $81.39_{\pm 2.54}$ | $65.95_{\pm 4.06}$ | $60.06_{\pm 3.21}$ | $85.46_{\pm 9.23}$ | $65.34_{\pm 5.57}$ | $66.18_{\pm 5.22}$ |
| | Edge Dropping | $81.29_{\pm 1.57}$ | $66.03_{\pm 4.06}$ | $59.86_{\pm 3.66}$ | $88.31_{\pm 5.95}$ | $66.98_{\pm 3.97}$ | $66.09_{\pm 4.81}$ |
| | Node Dropping | $81.33_{\pm 1.84}$ | $65.58_{\pm 3.46}$ | $60.55_{\pm 2.78}$ | $88.06_{\pm 7.20}$ | $65.78_{\pm 4.50}$ | $67.00_{\pm 5.69}$ |
| | Feature Dropping | $81.12_{\pm 1.78}$ | $65.87_{\pm 3.33}$ | $60.45_{\pm 2.36}$ | $85.51_{\pm 9.41}$ | $67.07_{\pm 4.86}$ | $62.64_{\pm 4.21}$ |
| | Mixup | $70.03_{\pm 3.99}$ | $50.19_{\pm 0.31}$ | $51.26_{\pm 1.16}$ | $70.29_{\pm 0.00}$ | $54.18_{\pm 3.26}$ | $68.53_{\pm 0.88}$ |
| | $\text{Aug}_{GE}$ | $\underline{82.45}_{\pm 1.18}$ | $\underline{66.52}_{\pm 1.55}$ | $\underline{64.80}_{\pm 1.28}$ | $\mathbf{95.09}_{\pm \mathbf{0.98}}$ | $\mathbf{72.33}_{\pm \mathbf{3.71}}$ | $\underline{68.09}_{\pm 4.09}$ |
| | $\text{Aug}_{PE}$ | $\mathbf{83.16}_{\pm \mathbf{2.10}}$ | $\mathbf{66.55}_{\pm \mathbf{1.31}}$ | $\mathbf{65.10}_{\pm \mathbf{1.39}}$ | $\mathbf{95.09}_{\pm \mathbf{0.98}}$ | $\underline{69.22}_{\pm 4.28}$ | $\mathbf{68.91}_{\pm \mathbf{4.18}}$ |
| PNA | Vanilla | $83.67_{\pm 4.78}$ | $73.74_{\pm 4.57}$ | $60.76_{\pm 4.57}$ | $87.77_{\pm 9.73}$ | $62.07_{\pm 3.60}$ | $67.18_{\pm 3.76}$ |
| | Edge Inserting | $82.07_{\pm 2.68}$ | $75.66_{\pm 2.02}$ | $59.55_{\pm 3.35}$ | $89.00_{\pm 4.40}$ | $\underline{64.22}_{\pm 4.58}$ | $65.00_{\pm 5.71}$ |
| | Edge Dropping | $82.48_{\pm 2.97}$ | $74.75_{\pm 2.46}$ | $59.15_{\pm 3.71}$ | $91.74_{\pm 3.02}$ | $\underline{64.22}_{\pm 5.76}$ | $62.00_{\pm 8.01}$ |
| | Node Dropping | $82.45_{\pm 2.36}$ | $75.11_{\pm 1.62}$ | $58.72_{\pm 3.38}$ | $91.37_{\pm 2.54}$ | $61.81_{\pm 7.44}$ | $63.91_{\pm 8.29}$ |
| | Feature Dropping | $82.65_{\pm 3.27}$ | $75.60_{\pm 2.22}$ | $61.20_{\pm 4.55}$ | $91.03_{\pm 2.43}$ | $65.26_{\pm 3.56}$ | $64.64_{\pm 6.28}$ |
| | Mixup | $50.00_{\pm 0.00}$ | $70.57_{\pm 1.30}$ | $55.30_{\pm 4.20}$ | $38.29_{\pm 0.00}$ | $55.45_{\pm 2.73}$ | $50.52_{\pm 3.37}$ |
| | $\text{Aug}_{GE}$ | $\mathbf{84.39}_{\pm \mathbf{1.55}}$ | $\underline{77.91}_{\pm 1.54}$ | $\mathbf{67.32}_{\pm \mathbf{2.68}}$ | $\mathbf{94.60}_{\pm \mathbf{1.10}}$ | $63.10_{\pm 3.65}$ | $\mathbf{69.36}_{\pm \mathbf{4.90}}$ |
| | $\text{Aug}_{PE}$ | $\underline{84.35}_{\pm 1.32}$ | $\mathbf{78.12}_{\pm \mathbf{1.37}}$ | $\underline{66.41}_{\pm 2.43}$ | $\underline{93.74}_{\pm 1.12}$ | $\mathbf{66.03}_{\pm \mathbf{3.66}}$ | $\underline{68.55}_{\pm 6.03}$ |
| GraphSage | Vanilla | $86.09_{\pm 2.62}$ | $62.58_{\pm 4.76}$ | $57.19_{\pm 4.87}$ | $92.60_{\pm 5.85}$ | $63.45_{\pm 4.79}$ | $66.73_{\pm 5.37}$ |
| | Edge Inserting | $85.17_{\pm 2.53}$ | $66.57_{\pm 3.94}$ | $55.01_{\pm 4.26}$ | $94.77_{\pm 1.28}$ | $65.95_{\pm 5.76}$ | $63.73_{\pm 5.46}$ |
| | Edge Dropping | $85.03_{\pm 2.69}$ | $66.43_{\pm 3.67}$ | $55.30_{\pm 3.89}$ | $94.69_{\pm 1.35}$ | $67.16_{\pm 2.71}$ | $63.82_{\pm 5.23}$ |
| | Node Dropping | $84.35_{\pm 2.58}$ | $\underline{67.67}_{\pm 3.56}$ | $55.18_{\pm 3.63}$ | $94.00_{\pm 2.08}$ | $66.98_{\pm 4.51}$ | $63.36_{\pm 4.69}$ |
| | Feature Dropping | $85.10_{\pm 2.28}$ | $66.37_{\pm 4.36}$ | $54.71_{\pm 3.63}$ | $93.97_{\pm 2.57}$ | $65.17_{\pm 4.31}$ | $62.09_{\pm 5.11}$ |
| | Mixup | $57.14_{\pm 0.00}$ | $50.00_{\pm 0.00}$ | $50.00_{\pm 0.00}$ | $61.71_{\pm 0.00}$ | $51.64_{\pm 1.12}$ | $40.00_{\pm 0.00}$ |
| | $\text{Aug}_{GE}$ | $\mathbf{88.50}_{\pm \mathbf{1.55}}$ | $67.69_{\pm 1.29}$ | $\underline{60.64}_{\pm 1.47}$ | $\underline{95.20}_{\pm 1.33}$ | $\underline{68.10}_{\pm 4.51}$ | $\mathbf{70.09}_{\pm \mathbf{5.35}}$ |
| | $\text{Aug}_{PE}$ | $\underline{87.86}_{\pm 2.39}$ | $\mathbf{67.86}_{\pm \mathbf{1.47}}$ | $\mathbf{63.38}_{\pm \mathbf{1.09}}$ | $\mathbf{95.23}_{\pm \mathbf{1.33}}$ | $\mathbf{68.28}_{\pm \mathbf{4.41}}$ | $\underline{70.00}_{\pm 5.30}$ |

Table 6: Performance comparisons with invariant methods 3-layer GCN trained on 50 samples. The metric is classification accuracy. The best results are shown in bold font and the second best ones are underlined.

| Dataset | MUTAG | Benzene | Fluoride | Alkane | D&D | PROTEINS |
|---|---|---|---|---|---|---|
| Vanilla | $84.29_{\pm 3.18}$ | $73.86_{\pm 5.20}$ | $62.07_{\pm 4.32}$ | $93.66_{\pm 3.22}$ | $63.19_{\pm 6.55}$ | $68.09_{\pm 6.12}$ |
| GSAT | $85.44_{\pm 2.52}$ | $60.99_{\pm 6.19}$ | $58.96_{\pm 5.87}$ | $89.46_{\pm 12.21}$ | $64.14_{\pm 6.58}$ | $68.27_{\pm 6.25}$ |
| DIR | $65.54_{\pm 4.56}$ | $64.47_{\pm 12.22}$ | $58.71_{\pm 6.13}$ | $76.46_{\pm 16.84}$ | $\mathbf{69.91}_{\pm \mathbf{5.50}}$ | $55.64_{\pm 3.79}$ |
| GALA | $65.31_{\pm 7.89}$ | $56.90_{\pm 5.53}$ | $54.16_{\pm 3.92}$ | $61.77_{\pm 18.65}$ | $60.34_{\pm 3.45}$ | $58.91_{\pm 7.11}$ |
| $\text{Aug}_{GE}$ | $\underline{87.17}_{\pm 1.44}$ | $\underline{76.20}_{\pm 1.31}$ | $\mathbf{66.55}_{\pm \mathbf{3.44}}$ | $\underline{96.31}_{\pm 1.29}$ | $66.12_{\pm 5.14}$ | $\underline{70.45}_{\pm 5.90}$ |
| $\text{Aug}_{PE}$ | $\mathbf{87.24}_{\pm \mathbf{2.56}}$ | $\mathbf{76.52}_{\pm \mathbf{0.76}}$ | $\underline{65.33}_{\pm 4.96}$ | $\mathbf{96.43}_{\pm \mathbf{1.12}}$ | $\underline{67.67}_{\pm 4.31}$ | $\mathbf{71.18}_{\pm \mathbf{6.34}}$ |

