# OpenReview forum: "Explanation-Assisted Data Augmentation  for Graph Learning"
_ICLR.cc/2025/Conference — ICLR 2025 Conference Withdrawn Submission_

### Official Review · Reviewer_uX2Z · 2024-10-31

**Soundness:** 3
**Presentation:** 3
**Contribution:** 3
**Rating:** 6
**Confidence:** 2

**Summary:**

This paper proposes propose explanation-assisted DA (EA-DA) for Graph Neural Networks (GNNs). The central idea is to leverage subgraph explanations, with 'almost sufficient' statistics for the classification label, to create label-preserving graph perturbations. The authors present some theoretical analysis of EA-DA, and propose an explanation-assisted empirical risk minimization (EA-ERM) learning rule. The empirical evaluations are also provided demonstrating the effectiveness of EA-DA in improving the performance of Graph Neural Networks (GNNs).

**Strengths:**

1.	The paper provides a rigorous theoretical analysis for EA-DA, including the derivation of sample complexity bounds.
2.	The concept of using subgraph explanations to guide data augmentation is innovative.
3.	The authors support their theoretical claims with extensive empirical evaluations, demonstrating the practical benefits of EA-DA across various datasets and GNN architectures.

**Weaknesses:**

1.	It appears the effectiveness of EA-DA is heavily dependent on the quality of the subgraph explanations. If the explanations are not accurate, the augmented samples may not be label-preserving.
2.	The EA-DA approach may be more complicated to implement compared to traditional DA techniques. The justification of the practical usefulness of the proposed methods is needed.

**Questions:**

1.	How does EA-DA perform in dynamic graph settings where the graph structure changes over time?
2.	What are the computational costs associated with generating explanation-assisted augmentations, and how do they scale with graph size?
3.	Could the authors provide more insight into how EA-DA balances the trade-off between in-distribution and out-of-distribution augmentations?

---

### Official Review · Reviewer_GHnz · 2024-11-01

**Soundness:** 3
**Presentation:** 2
**Contribution:** 3
**Rating:** 5
**Confidence:** 2

**Summary:**

This paper focuses on the research problem of data augmentations on graph-based data and proposes explanation-assisted DA (EA-DA), which enables label-preserving augmentations by employing perturbation on non-explanation subgraphs. The study also includes theoretically supported discussions on sample complexity and optimal use cases. Empirical experiments are presented to confirm the theoretical insights.

**Strengths:**

This paper provides rigorous mathematical notations and extensive theoretical discussions regarding data augmentations.

**Weaknesses:**

1. While the paper includes thorough theoretical discussions, it lacks specific details on the practical implementation of key components:
  - **Classifier $f(\cdot)$**: There is no description of the pre-training process.
  - **Explainer $\Psi(\cdot)$**: The authors mention that real-world applications may not have ground-truth explanations available and the explainer is pre-trained (lines 391- 392) in this paper. However, the author does not provide clear details on how this pre-training addresses such limitations. Additionally, important aspects of the explainer’s architecture, including model type and learning approach, are omitted. These design details are crucial to understanding the overall framework, just as much as the theoretical foundations.

2. In experimental settings, it is stated that GNNExplainer [1] and PGExplainer [2] are adopted as the **explainer $\Psi(\cdot)$** instead of designing a novel one, raising a few concerns:
  - It is uncertain whether GNNExplainer and PGExplainer align with the assumptions and theoretical conclusions presented in earlier sections.
  - Moreover, the potential increase in computational complexity, such as training time, due to EA-DA is unclear. An efficiency comparison would be helpful to determine whether EA-DA effectively balances performance and resource consumption.

**Questions:**

See Weaknesses.

---

### Official Review · Reviewer_JBdc · 2024-11-04

**Soundness:** 2
**Presentation:** 2
**Contribution:** 2
**Rating:** 5
**Confidence:** 2

**Summary:**

This paper presents explanation-assisted Data Augmentation (EA-DA) for Graph Neural Networks (GNNs), a novel technique designed for graph learning. EA-DA employs graph explanations—subgraphs that act as "almost sufficient" statistics—to maintain label invariance against edge perturbations. The findings indicate that EA-DA can significantly reduce sample complexity compared to explanation-agnostic methods, although failing to differentiate between augmented and original data may lead to increased complexity. Overall, the study theorizes that EA-DA can improve sample complexity with suitable learning mechanisms and offers practical implementations supported by empirical evaluations.

**Strengths:**

This paper formulates explanation-assisted Data Augmentation (EA-DA) mechanisms for graph learning and addresses sample complexity issues. It introduces the explanation-assisted empirical risk minimization (EA-ERM) learning rule, demonstrating that it can achieve significantly lower sample complexity than explanation-agnostic methods. The study shows that failing to distinguish between original and augmented samples in EA-DA may worsen sample complexity. Additionally, an implementable class of EA-DA mechanisms is presented, along with empirical simulations that confirm improved performance of GNNs trained with EA-DA under suitable conditions, as well as scenarios where these conditions are unmet, highlighting the practical benefits of the approach

**Weaknesses:**

The experimental section of this paper does not compare with many existing articles on graph data augmentation, which I believe is a significant oversight and a key issue. The list of articles is as follows:

[1] LSPAN: Spectrally Localized Augmentation for Graph Consistency Learning.
[2] Graph random neural networks for semi-supervised learning on graphs.
[3] Spectral augmentation for self-supervised learning on graphs.
[4] Dropedge: Towards deep graph convolutional networks on node classifcation.
[5]Data augmentation for graph neural networks.

In terms of datasets, commonly used graph datasets such as CORA, CITESEER, PPI, BLOGC, FLICKR, and AIR-USA were also not considered, which I believe undermines the persuasiveness of the study.

**Questions:**

See the weaknesses.

---

### Official Review · Reviewer_oC7Q · 2024-11-04

**Soundness:** 3
**Presentation:** 3
**Contribution:** 2
**Rating:** 5
**Confidence:** 4

**Summary:**

The paper introduces Explanation-Assisted Data Augmentation (EA-DA) for GNNs a novel data augmentation technique that leverages graph explanations to improve the robustness and generalization of GNNs. The authors provide theoretical insights into the sample complexity of EA-DA, as well as empirical validation of the method's effectiveness on various graph learning tasks, demonstrating its superior performance over existing augmentation techniques.

**Strengths:**

This paper presents a novel approach to data augmentation in the context of graph learning by introducing the concept of using explanation subgraphs. This is an innovative integration of explainability techniques with data augmentation. Overall, the paper is well-structured and clearly written.

**Weaknesses:**

1. One potential limitation of EA-DA is its reliance on the quality of the explanation subgraphs. The method assumes that the explanation subgraph effectively represents the most important parts of the graph for classification. However, existing graph explanation methods (e.g., GNNExplainer, PGExplainer) are not perfect, and their outputs may be noisy or incomplete. If the explanation subgraphs are inaccurate, the augmented data might lead to performance degradation. The paper could benefit from a sensitivity analysis showing how robust the method is to low-quality explanations.

2. While the method focuses on perturbing non-explanatory edges, the diversity of the augmented samples may be limited. The current perturbation strategy modifies the graph structure only by adding or removing edges, which may not fully explore the potential augmentation space.

3. The process of generating explanation subgraphs for each sample could be computationally expensive, especially for large graphs or large datasets. Although the authors do not explicitly discuss the computational cost of their method, this could be a practical limitation in real-world scenarios where efficiency is critical. A discussion of the trade-offs between the increased computational complexity of EA-DA and the performance gains it offers would be useful.

4. The paper demonstrates the effectiveness of EA-DA on several benchmark datasets, but it is unclear how well this method would generalize to very large-scale graphs, dynamic graphs, or graphs with different structural properties (e.g., dense graphs). A discussion or analysis of the method's scalability and applicability to a broader range of tasks would strengthen the paper's argument.

**Questions:**

1. **Explanation Quality**:
   How sensitive is EA-DA to the quality of the explanation subgraphs? Have the authors experimented with different explanation methods, and if so, how does the choice of explanation technique impact the performance of the method?

2. **Augmentation Strategy**:
   Why did the authors choose to restrict the augmentation to edge perturbations, and did they explore other types of augmentations (e.g., modifying node features, changing graph structure)? Could more diverse augmentation strategies improve the model's robustness further?

3. **Computational Complexity**:
   Can the authors provide more details on the computational requirements of EA-DA? Specifically, how does the time complexity of generating explanation subgraphs and performing augmentations compare to the baseline methods? Would this approach be feasible for very large graphs?

---

### Official Review · Reviewer_axc5 · 2024-11-04

**Soundness:** 3
**Presentation:** 3
**Contribution:** 3
**Rating:** 6
**Confidence:** 3

**Summary:**

This paper proposes a data augmentation technique for graphs, termed explanation-assisted data augmentation, which leverages a function explainer map. The core idea is to generate perturbations of the graph that preserve the classification label by focusing on edges or nodes not included in the explanation subgraph, thus augmenting the dataset with explanation-preserving perturbations. The classifier is retrained on this augmented dataset with a loss function that differentiates between original and augmented data.

The main focus of the paper is on the theoretical aspects of explanation-assisted learning. It introduces the concepts of explanation-assisted sample complexity, empirical risk minimization, and VC dimension. The authors show that the sample complexity of explanation-assisted learning can be significantly smaller than that of explanation-agnostic learning. However, if the learning algorithm does not distinguish between augmented and original data, the sample complexity can actually worsen due to the out-of-distribution nature of the graph perturbations. The paper does not detail the specific explainer or the exact perturbation methods but instead provides a theoretical framework and empirical evaluations to support the proposed EA-DA techniques.

**Strengths:**

- The paper is well-organized and clearly presented.

- The contributions are novel, particularly in introducing explanation-assisted data augmentation to graph learning and performing an analizing it with respect to the PAC learning framework.

**Weaknesses:**

- The method necessitates training the model twice, adding to the computational cost. Specifically, it requires an initial pretraining phase on the original dataset to obtain a functional model, followed by training an explainer on this pretrained model. After generating the augmented dataset based on this explainer, the model is then retrained on both the original and augmented datasets, effectively doubling the training effort.

- The augmentation technique is highly model-dependent, meaning that the generated augmentations may not be transferable to other models, or at least this option is not discussed in the paper.

Minor:
- Including references to recent relevant works, such as GeoMIX [1], and adding recent methods like [2] to Table 1 would enhance the paper by providing a more comprehensive comparison with existing approaches.

[1] Zhao, Wentao, et al. "GeoMix: Towards Geometry-Aware Data Augmentation." Proceedings of the 30th ACM SIGKDD Conference on Knowledge Discovery and Data Mining. 2024.
[2] Ling, Hongyi, et al. "Graph mixup with soft alignments." International Conference on Machine Learning. PMLR, 2023.

**Questions:**

- The experiments and overall discussion are focused on graph classification. Do you believe this approach could be extended to other tasks, such as node classification or edge prediction?
- How can we assess whether the generated data points are out-of-distribution?
- What motivated the decision to run the experiments in Table 1 using only 50 training samples?
- Should $\Pi$ and $\Psi$ be provided as inputs to Algorithm 1?
- It would be useful to report information related to running time, even acknowledging the dependency on specific modules. Could you provide some insight into this aspect?

---

### Note · Authors · 2024-11-28

I have read and agree with the venue's withdrawal policy on behalf of myself and my co-authors.